# Unraveling the mechanistic links between blood pressure regulation and calcium-magnesium homeostasis: Insights into hypertension, hyperparathyroidism, and mineral disorders

Pritha Dutta[1], Anita T. Layton [2]*

**1** Department of Applied Mathematics, University of Waterloo, Waterloo, Ontario, Canada, **2** Departments of Applied Mathematics and Biology, School of Pharmacy, Cheriton School of Computer Science, University of Waterloo, Waterloo, Ontario, Canada

* anita.layton@uwaterloo.ca

## Abstract

The systems regulating blood pressure and calcium-magnesium ($Ca^{2+}$-$Mg^{2+}$) homeostasis are increasingly recognized to have clinically relevant interactions, where alterations in one can lead to significant changes in the other. In this study, we developed a computational model integrating blood pressure regulation and $Ca^{2+}$-$Mg^{2+}$ homeostasis in a male rat. We simulated various conditions, including hypertension, $Ca^{2+}$, $Mg^{2+}$, and vitamin $D_3$ deficiencies, and primary hyperparathyroidism. Simulations of hypertension, induced by various stimuli like increased renin or aldosterone secretion, demonstrated significant effects on parathyroid hormone (PTH), calcitriol, renal $Ca^{2+}$/$Mg^{2+}$ handling, and bone resorption. Dietary $Ca^{2+}$, $Mg^{2+}$, and vitamin $D_3$ deficiencies was predicted to elevate mean arterial pressure, with $Mg^{2+}$ deficiency having a stronger effect. Furthermore, the model predicted that primary hyperparathyroidism elevates PTH, $Ca^{2+}$, and calcitriol, leading to increased mean arterial pressure and bone loss. Overall, this model provides valuable insights into the mechanistic links between blood pressure regulation and $Ca^{2+}$-$Mg^{2+}$ homeostasis, offering insights into clinical conditions like hypertension and hyperparathyroidism.

## Introduction

Hypertension is the leading cause of cardiovascular disease worldwide [1], and its prevalence has been rising in recent years due to factors such as an aging population and lifestyle changes, including decreased physical activity and the widespread consumption of Western diets [2]. Furthermore, a significant portion of the global population fails to meet the recommended dietary intake of magnesium ($Mg^{2+}$) and calcium ($Ca^{2+}$) [3,4]. For example, the standard diet in the United States provides only about 50% of the recommended daily intake of $Mg^{2+}$ [5], and the 2017-2018 National Health and Nutrition Examination Survey found that nearly half of the United States

**Data availability statement:** Data are available at

https://github.com/Pritha17/Blood-pressure-and-calcium-magnesium-homeostasis-regulation.

**Funding:** This research was supported in part by the Natural Sciences and Engineering Research Council of Canada, via a Discovery award RGPIN-2025-03958 to AT Layton. The funders had no role in study design, data collection and analysis, decision to publish, or preparation of the manuscript.

**Competing interests:** The authors have declared that no competing interests exist.

population did not meet their estimated average requirements for $Ca^{2+}$ [6]. Additionally, approximately 100,000 people in the United States each year are diagnosed with primary hyperparathyroidism [7], which involves elevated blood $Ca^{2+}$ and its regulatory hormone parathyroid hormone (PTH) levels. Given these factors, it is essential to understand how blood pressure regulation and $Ca^{2+}$-$Mg^{2+}$ homeostasis are affected by these conditions.

The systems regulating blood pressure and $Ca^{2+}$-$Mg^{2+}$ homeostasis are increasingly recognized to have significant and clinically relevant interactions [8–10]. A key component of blood pressure regulation is the renin-angiotensin-aldosterone system (RAAS). The RAAS is a hormonal system that regulates fluid balance and blood pressure through a cascade of reactions. One of the end products angiotensin II (Ang II) promotes vascular constriction and stimulates the adrenal glands to release aldosterone, which enhances sodium ($Na^+$) and water retention by the kidneys, ultimately increasing blood volume and pressure. $Ca^{2+}$, $Mg^{2+}$, and their regulatory hormones, PTH and calcitriol ($1,25(OH)_2D_3$; active form of vitamin $D_3$), play a crucial role in regulating various elements of the RAAS [8–10]. For instance, $Ca^{2+}$ and calcitriol influence renin secretion from the juxtaglomerular (JG) cells in the kidneys [11–14]. Additionally, $Ca^{2+}$, $Mg^{2+}$, and PTH regulate aldosterone secretion from the adrenal gland [15–19] as well as vascular resistance [20–22]. Both aldosterone and Ang II also influence PTH secretion from the parathyroid gland [23]. Furthermore, PTH and $Ca^{2+}$ control $Na^+$ reabsorption of the kidney, specifically, along the proximal tubule and thick ascending limb segments of the nephron [24,25]; $Na^+$ reabsorption in turn affects the reabsorption of $Ca^{2+}$ and $Mg^{2+}$ in these nephron segments. Ang II and aldosterone also indirectly influence $Ca^{2+}$ and $Mg^{2+}$ reabsorption in the kidneys by regulating $Na^+$ reabsorption. Finally, $Ca^{2+}$ and $Mg^{2+}$ have opposing effects on renal sympathetic nervous activity (RSNA) [26,27], which modulates renin secretion and renal $Na^+$ transport. This complex network of interconnections between the two systems is illustrated by the brown arrows in Fig 1. As a result of these couplings and feedbacks, dysregulation in one system can significantly impact the other. Therefore, understanding the interactions between these systems is increasingly important for accurately assessing their impact on health.

Mechanistic modeling provides a comprehensive framework for understanding and analyzing complex physiological systems. In this study, we integrated our previously developed $Ca^{2+}$-$Mg^{2+}$ homeostasis model [28] with a blood pressure regulation model [29], both formulated for the male rat, to quantify the interactions between these two systems. This integrated model was then used to simulate various conditions, including hypertension, $Ca^{2+}$, $Mg^{2+}$, and vitamin D deficiencies, and primary hyperparathyroidism.

## Materials and methods

As noted above, the present model integrates our previously developed model of $Ca^{2+}$ and $Mg^{2+}$ homeostasis [28] with a blood pressure regulation model [29]. The blood pressure regulation model describes, using a large system of coupled nonlinear algebraic differential equations, the interactions among physiological systems

key to this process: the cardiovascular system, the renal system, the renal sympathetic nervous system, and the RAAS. The latter three regulate blood pressure via their effects on vascular resistance and on electrolyte and fluid balance. A schematic diagram is shown in Fig 2A. Model equations and parameter values can be found in Ref. [29].

The $Ca^{2+}$ and $Mg^{2+}$ homeostasis model consists of five compartments: plasma, parathyroid gland, intestine, kidney, and bone. The model describes, using a large system of coupled nonlinear ordinary differential equations, the exchanges of $Ca^{2+}$, $Mg^{2+}$, PTH, and calcitriol among these compartments. These exchanges together determine plasma $[Ca^{2+}]$ and $[Mg^{2+}]$, which are kept within physiological ranges by means of feedback controls. A schematic diagram is shown in Fig 2B. Model equations and parameter values can be found in Ref. [28].

In the following sections we define the equations for the regulation of different components of the blood pressure regulation model by $Ca^{2+}$, $Mg^{2+}$, PTH, and $1,25(OH)_2D_3$ and vice-versa. As previously noted, the interactions between the two systems are summarized in Fig 1. New parameters that characterize these interactions are listed in Table 1.

## RAAS: Renin secretion

We begin with the RAAS, which as noted above is a key hormonal system that regulates fluid balance, primarily via its actions on the kidney, and vascular tone through a series of biochemical reactions. The reaction cascade "begins" with

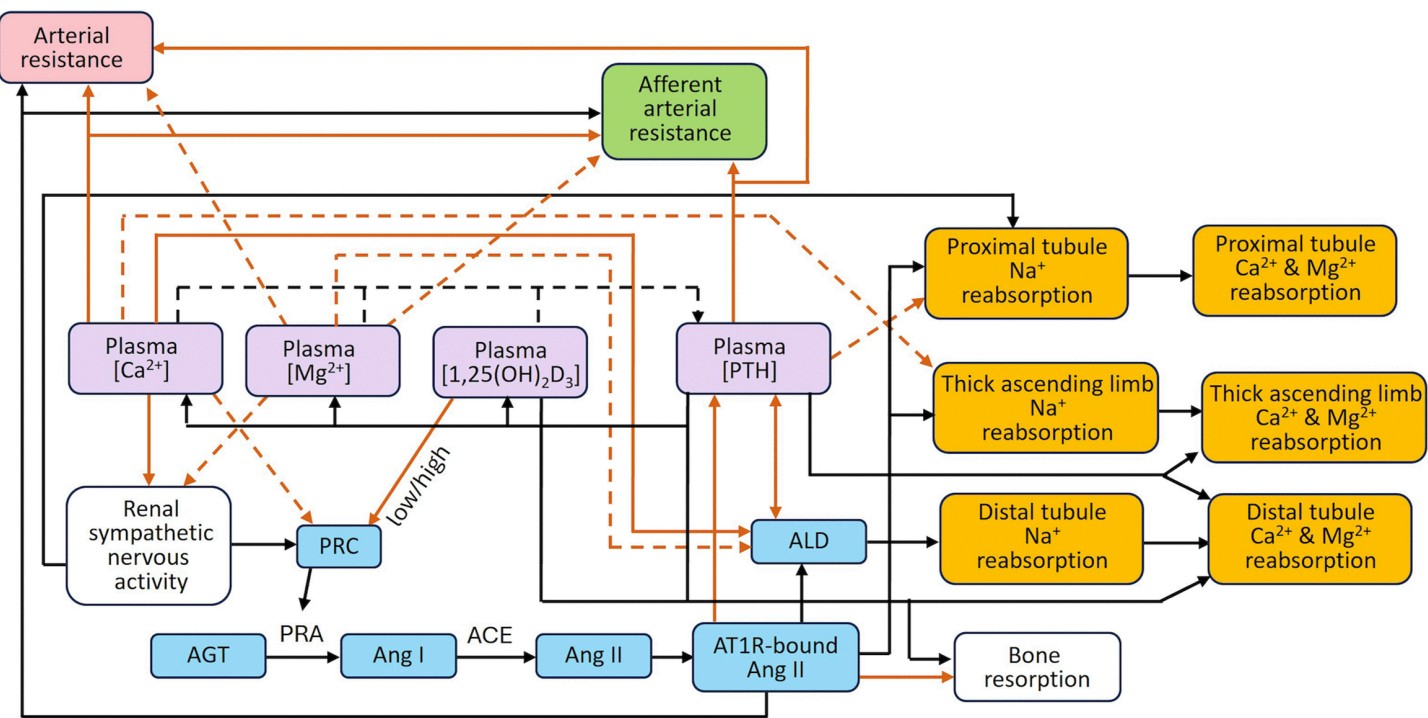

**Fig 1. Schematic representation of the interactions between the blood pressure and $Ca^{2+}$-$Mg^{2+}$ homeostasis models.** Mauve nodes, plasma $[Ca^{2+}]$, $[Mg^{2+}]$, $[1,25(OH)_2D_3]$, and [PTH]; yellow nodes, renal $Na^+$, $Ca^{2+}$, and $Mg^{2+}$ handling; blue nodes, renin-angiotensin-aldosterone system. (Node colors match those in Fig 2.) Solid arrows indicate activation and dotted arrows indicate inhibition. Brown arrows indicate the direct links between components of the blood pressure regulation model and the $Ca^{2+}$-$Mg^{2+}$ homeostasis model. In this schematic, distal tubule includes the distal convoluted tubule, connecting tubule, and collecting duct. ACE, angiotensin-converting enzyme; AGT, angiotensinogen; ALD, aldosterone; Ang I, angiotensin I; Ang II, angiotensin II; AT1R-bound Ang II, angiotensin II type 1 receptor-bound angiotensin II; AT2R-bound Ang II, angiotensin II type 2 receptor-bound angiotensin II; PRC, plasma renin concentration.

**(A)**

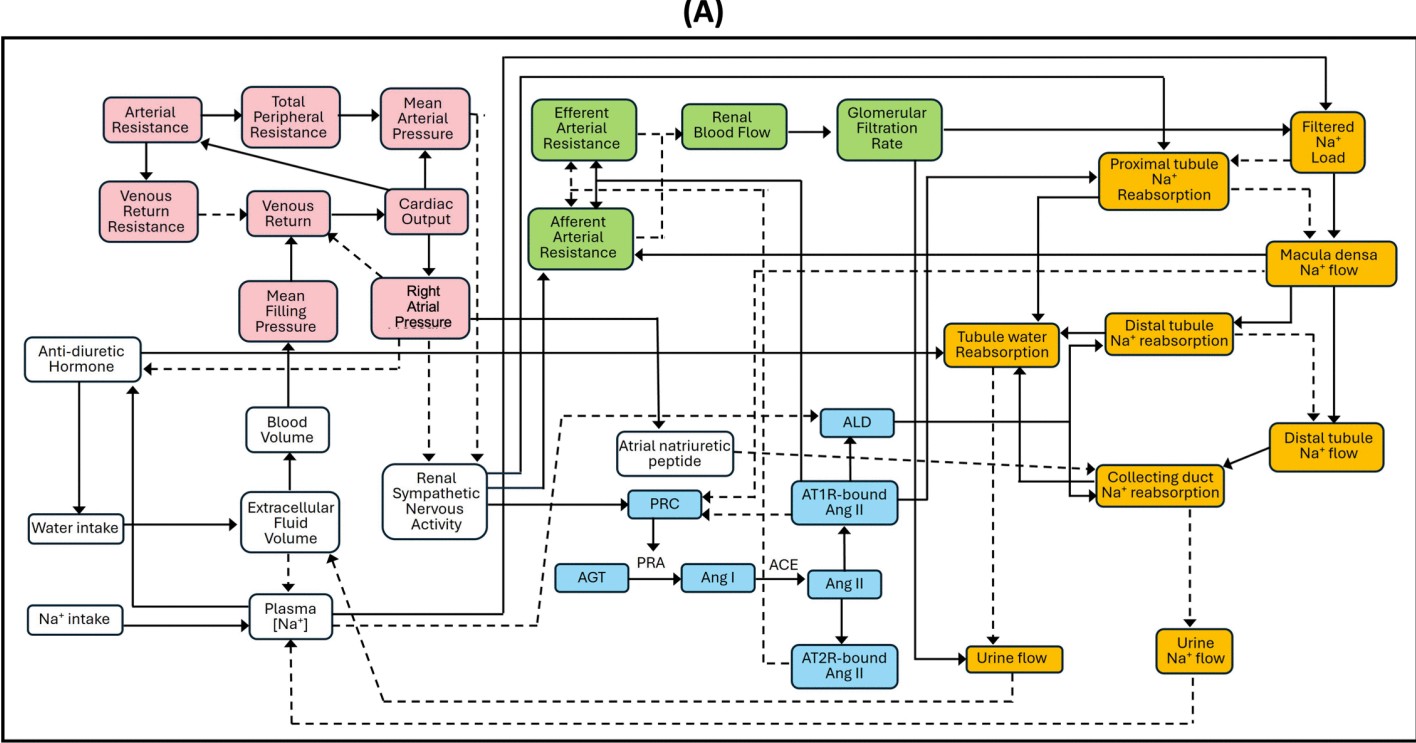

**(B)**

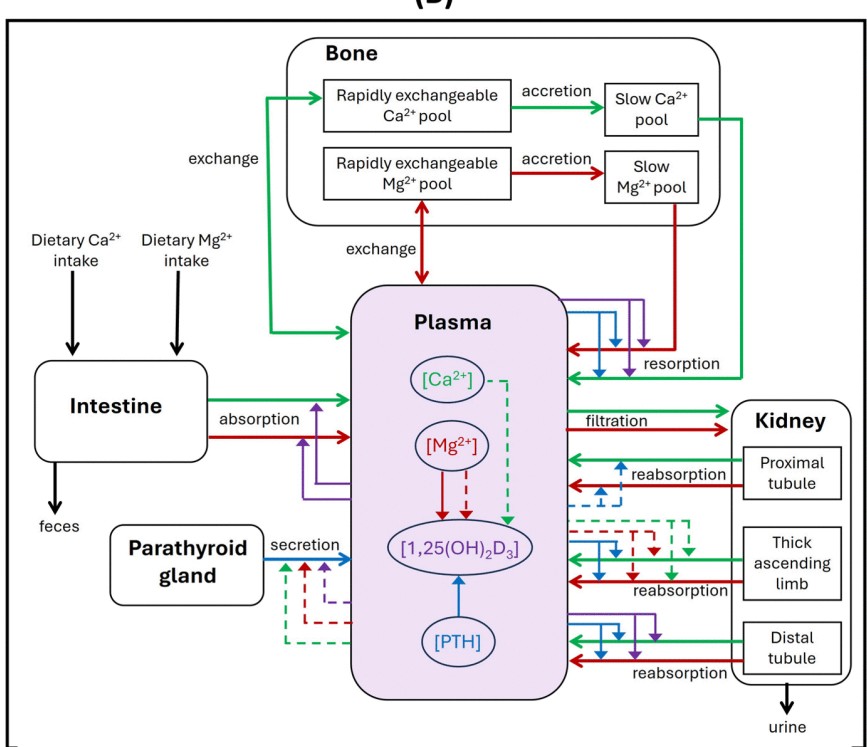

**Fig 2**. **Schematic diagram of the integrative model.** (A) Blood pressure regulation model. Pink nodes denote variables that describe cardiovascular function; green nodes, renal hemodynamics; yellow nodes, renal Na$^+$ and fluid handling; blue nodes, the renin-angiotensin-aldosterone system (RAAS). Solid arrows indicate activation and dotted arrows indicate inhibition. In this schematic, the proximal tubule includes also the loop of Henle. (B) Ca$^{2+}$-Mg$^{2+}$ homeostasis model. The model consists of five compartments: plasma, intestine, kidney, parathyroid gland, and bone. Solid arrows

with open arrowheads indicate fluxes, solid arrows with closed arrowheads indicate activation, and dotted arrows indicate inhibition. All arrows are color coded. Green arrows, $Ca^{2+}$; red arrows, $Mg^{2+}$, blue arrows, parathyroid hormone (PTH); purple arrows, $1,25(OH)_2D_3$. ACE, angiotensin-converting enzyme; AGT, angiotensinogen; ALD, aldosterone; Ang I, angiotensin I; Ang II, angiotensin II; AT1R-bound Ang II, angiotensin II type 1 receptor-bound angiotensin II; AT2R-bound Ang II, angiotensin II type 2 receptor-bound angiotensin II; PRC, plasma renin concentration.

**Table 1**. Descriptions and values of new model parameters and variables at baseline for a male rat.

| Symbol | Description | Value | Ref. | Comment |
|---|---|---|---|---|
| $K_{R\text{-}Ca}$ | Sensitivity of renin secretion to plasma $Ca^{2+}$ concentration | 5.4 mM | [11] | rat study |
| $\nu_{Ca}$ | Effect of $Cca^{2+}$ on renin secretion | 1 | | |
| $K_{R-1,25(OH)_2D_3-low}$ | Sensitivity of renin secretion to low plasma $1,25(OH)_2D_3$ concentration | 30 pM | [13] | mouse study |
| $K_{R-1,25(OH)_2D_3-high}$ | Sensitivity of renin secretion to high plasma $1,25(OH)_2D_3$ concentration | 200 pM | [14] | mouse study |
| $\nu_{1,25(OH)_2D_3}$ | Effect of $1,25(OH)_2D_3$ on renin secretion | 1 | | |
| $K_{ALD\text{-}Ca}$ | Sensitivity of aldosterone secretion to plasma Ca concentration | 2.67 mM | [15] | rat study |
| $\xi_{Mg}$ | Effect of $Mg^{2+}$ on aldosterone secretion | 1 | | |
| $K_{ALD\text{-}PTH}$ | Sensitivity of aldosterone secretion to plasma PTH concentration | 124 pM | [39] | rat study |
| $\xi_{PTH}$ | Effect of PTH on aldosterone secretion | 1 | | |
| $K_{PTH\text{-}ALD}$ | Sensitivity of PTH secretion to aldosterone concentration | 800 ng/l | [23] | human parathyroid cell study |
| $K_{PTH\text{-}AT1R}$ | Sensitivity of PTH secretion to AT1R-bound Ang II concentration | 42 fmol/l | [23] | human parathyroid cell study |
| $\alpha_{Ca}$ | Effect of $Ca^{2+}$ on RSAN | 1 | | |
| $\alpha_{Mg}$ | Effect of $Mg^{2+}$ on RSAN | 1 | | |
| $K_{res}^{AT1R}$ | Sensitivity of bone resorption to AT1R-bound Ang II | 20 fmol/ml | | Setl to equilibrium [AT1R-bound Ang II] |
| $K_{sod\text{-}PTH}$ | Sensitivity of $Na^+$ reabsorption in proximal tubule to PTH | 2.39 pM | | |
| $\gamma_{PTH}$ | Effect of PTH on fractional $Na^+$ reabsorption in the proximal tubule | 1 | | |
| $K_{sod\text{-}Ca}$ | Sensitivity of $Na^+$ reabsorption in proximal tubule to $Ca^{2+}$ | 1.25 mM | | |
| $\gamma_{Ca}$ | Effect of $Ca^{2+}$ on fractional $Na^+$ reabsorption in the proximal tubule | 1 | | |
| $\psi_{AT1R}$ | Effect of AT1R-bound Ang II on arterial resistance | 1 | | |
| $\psi_{Ca}$ | Effect of $Ca^{2+}$ on arterial resistance | 1 | | |
| $\psi_{Mg}$ | Effect of $Mg^{2+}$ on arterial resistance | 1 | | |
| $\psi_{PTH}$ | Effect of PTH on arterial resistance | 1 | | |

renin, which is secreted by the JG cells in the kidneys. Renin secretion is given by

$$R_{sec} = N_{rs} \times \nu_{md\text{-}sod} \times \nu_{rsna} \times \nu_{AT1R} \times \nu_{Ca} \times \nu_{1,25(OH)_2D_3} \tag{1}$$

where $N_{rs}$ represents normalized renin secretion, and $\nu_{md\text{-}sod}$, $\nu_{rsna}$, $\nu_{AT1R}$, $\nu_{Ca}$, and $\nu_{(1,25(OH)_2D_3}$ represent the effect of tubular $Na^+$ flow past the macula densa, RSNA, Ang II receptor type 1 (AT1R)-bound Ang II concentration, plasma $Ca^{2+}$ ($[Ca^{2+}]_p$), and plasma $1,25(OH)_2D_3$ concentration ($[1,25(OH)_2D_3]_p$) on renin secretion rate. All these terms are equal to 1 in the baseline condition. For definitions of $N_{rs}$, $\nu_{md\text{-}sod}$, $\nu_{rsna}$, and $\nu_{AT1R}$ refer to Ref. [29]. The terms $\nu_{Ca}$, and $\nu_{(1,25(OH)_2D_3}$ are introduced in this study and defined below.

Although $Ca^{2+}$ does not directly regulate renin secretion, when plasma $Ca^{2+}$ concentration is elevated, cAMP-stimulated renin secretion is inhibited, and vice versa (see dashed brown arrow between the plasma $[Ca^{2+}]$ and PRC nodes in Fig 1). Acute hypercalcemia suppresses plasma renin activity (PRA) by acting on the calcium-sensing receptor (CaSR) [11]. The JG cells, which express CaSR, reduce renin secretion when the receptor is activated [12]. The basolateral surface of the JG cells is exposed to the renal cortical interstitium, so elevated $Ca^{2+}$ levels in this area could directly stimulate the CaSR on JG cells, leading to a reduction in renin secretion. The effect of $Ca^{2+}$ on renin secretion is given by

$$\nu_{Ca} = \frac{(K_{R\text{-}Ca})^6}{(K_{R\text{-}Ca})^6 + ([Ca]_p)^6} \tag{2}$$

where $K_{\text{R-Ca}}$ represents the sensitivity of renin secretion to plasma [$Ca^{2+}$].

Renin secretion increases significantly in the presence of $1,25(OH)_2D_3$ deficiency as well as toxicity. Zhou et al. reported a 1.5-fold increase in PRA in $1\alpha$-hydroxylase knockout mice compared to control [13]. Treatment of $1\alpha$-hydroxylase knockout mice with $1,25(OH)_2D_3$ led to the normalization of PRA [13]. Thus, $1,25(OH)_2D_3$ deficiency leads to upregulation of renin secretion. In addition, administration of high doses of calcitriol to male rats increased PRA by $\sim$2.5-fold [14]. Thus, sufficiently high levels of $1,25(OH)_2D_3$ also upregulates renin secretion.

We defined $\nu_{1,25(OH)_2D_3}$ such that (i) when $[1,25(OH)_2D_3]_p$ is within the range of 80-170 pM, $\nu_{1,25(OH)_2D_3} \approx 1$, (ii) when $[1,25(OH)_2D_3]_p$ is less than 80 pM, $\nu_{1,25(OH)_2D_3}$ gradually increases to a maximum of 1.5-fold as $[1,25(OH)_2D_3]_p$ approaches zero, and (iii) when $[1,25(OH)_2D_3]_p$ is greater than 170 pM, $\nu_{1,25(OH)_2D_3}$ gradually increases to a maximum of 2-fold. Thus, the effect of $1,25(OH)_2D_3$ on renin secretion is given by

$$\nu_{1,25(OH)_2D_3} = (1 - h) \times \nu_{1,25(OH)_2D_3}^{\text{low,normal}} \times \nu_{1,25(OH)_2D_3}^{\text{high}} \tag{3}$$

where

$$h = \frac{\left([1,25(OH)_2D_3]_p/180 \text{ pM}\right)^{50}}{1 + \left([1,25(OH)_2D_3]_p/180 \text{ pM}\right)^{50}} \tag{4}$$

$$\nu_{1,25(OH)_2D_3}^{\text{low,normal}} = 0.99 + \frac{0.5 \times (K_{R-1,25(OH)_2D_3-\text{low}})^4}{(K_{R-1,25(OH)_2D_3-\text{low}})^4 + ([1,25(OH)_2D_3]_p)^4} \tag{5}$$

$$\nu_{1,25(OH)_2D_3}^{\text{high}} = 1 + \frac{([1,25(OH)_2D_3]_p)^4}{(K_{R-1,25(OH)_2D_3-\text{high}})^4 + ([1,25(OH)_2D_3]_p)^4} \tag{6}$$

The parameters $K_{R-1,25(OH)_2D_3-\text{low}}$ and $K_{R-1,25(OH)_2D_3-\text{high}}$ represent the sensitivity of renin secretion to low and high $[1,25(OH)_2 D_3]_p$, respectively.

## RAAS: Aldosterone secretion

Aldosterone is the end product of the RAAS reaction cascade. It elevates blood pressure via its $Na^+$ retention effect on the kidneys. Normalized aldosterone secretion ($N_{\text{als}}$) is given by

$$N_{\text{als}} = N_{\text{als}}^{\text{eq}} \times \xi_{\text{k/sod}} \times \xi_{\text{map}} \times \xi_{\text{AT1R}} \times \xi_{\text{Ca}} \times \xi_{\text{Mg}} \times \xi_{\text{PTH}} \tag{7}$$

ALD secretion is affected by the plasma $K^+$ to $Na^+$ ratio ($\xi_{\text{k/sod}}$), mean arterial pressure ($\xi_{\text{map}}$), AT1R-bound Ang II ($\xi_{\text{AT1R}}$), $Ca^{2+}$ ($\xi_{\text{Ca}}$), $Mg^{2+}$ ($\xi_{\text{Mg}}$), and PTH ($\xi_{\text{PTH}}$). All these terms are equal to 1 in the baseline condition. For definitions of $N_{\text{als}}^{\text{eq}}$, $\xi_{\text{k/sod}}$, $\xi_{\text{map}}$, and $\xi_{\text{AT1R}}$ refer to Ref. [29]. The terms $\xi_{\text{Ca}}$, $\xi_{\text{Mg}}$, and $\xi_{\text{PTH}}$ are defined below.

Findings in rat zona glomerulosa (ZG) cells [15] have suggested that extracellular $Ca^{2+}$ level could serve as an independent stimulator of aldosterone secretion. That effect is modelled as

$$\xi_{\text{Ca}} = 0.65 + \frac{([Ca^{2+}]_p)^{0.8}}{(K_{\text{ALD-Ca}})^{0.8} + ([Ca^{2+}]_p)^{0.8}} \tag{8}$$

where $K_{\text{ALD-Ca}}$ represents the sensitivity of aldosterone secretion to plasma $Ca^{2+}$ concentration.

Studies in cultured cell [16], rat [17], and human [30] have suggested that extracellular $Mg^{2+}$ decreases sensitivity of adrenal glomerulosa cells to Ang II-stimulated aldosterone secretion. We model the effect of $Mg^{2+}$ on aldosterone secretion as

$$\xi_{\text{Mg}} = \frac{1.57 \times (K_{\text{ALD-Mg}})^{1.2}}{(K_{\text{ALD-Mg}})^{1.2} + ([\text{Mg}^{2+}]_p)^{1.2}} \tag{9}$$

where $K_{\text{ALD-Mg}}$ represents the sensitivity of aldosterone secretion to plasma $\text{Mg}^{2+}$ concentration ($[\text{Mg}^{2+}]_p$).

PTH binds to PTH receptor and facilitates $\text{Ca}^{2+}$ influx through the voltage-gated $\text{Ca}^{2+}$ channels, thus increasing Ang II-stimulated aldosterone secretion [18,19,31,32]. That effect is modelled as

$$\xi_{\text{PTH}} = 0.9974 + \frac{([\text{PTH}]_p)^2}{(K_{\text{ALD-PTH}})^2 + ([\text{PTH}]_p)^2} \tag{10}$$

where $K_{\text{ALD-PTH}}$ represents the sensitivity of aldosterone secretion to $[\text{PTH}]_p$.

### Effect of angiotensin II and aldosterone on PTH secretion

Human parathyroid cells have been found to express mineralocorticoid receptors (MRs) [23,32] and AT1Rs [23]. The AT1R expression level was found to be ~100-fold lower than the MR expression level [23]. Lenzini et al. reported that at physiological $\text{Ca}^{2+}$ concentrations cells exposed to aldosterone increased PTH secretion by 271% compared to control [23]. This increase in PTH secretion was abolished on treatment with an MR blocker. In addition, at physiological $\text{Ca}^{2+}$ concentrations cells exposed to Ang II increased PTH secretion by 267% and this increase was abolished on treatment with an AT1R antagonist [23]. However, co-stimulation with both aldosterone and Ang II did not produce any additive increase in PTH secretion, with PTH secretion increasing by 225% compared to control [23]. Based on these observations, we model the effect of aldosterone and Ang II on PTH secretion as

$$F1(\text{ALD,AT1R}) = \min\left(2.7, 1 + F2(\text{ALD}) + F3(\text{AT1R})\right) \tag{11}$$

where

$$F2(\text{ALD}) = \frac{1.72 \times ([\text{ALD}]_p)^8}{(K_{\text{PTH-ALD}})^8 + ([\text{ALD}]_p)^8} \tag{12}$$

$$F3(\text{AT1R}) = \frac{1.62 \times [\text{AT1R}]^8}{(K_{\text{PTH-AT1R}})^8 + [\text{AT1R}]^8} \tag{13}$$

In Eqs (12) and (13), $[\text{ALD}]_p$ denotes plasma aldosterone concentration and $[\text{AT1R}]$ denotes AT1R-bound Ang II concentration. $K_{\text{PTH-ALD}}$ and $K_{\text{PTH-AT1R}}$ characterize the sensitivity of PTH secretion to aldosterone and AT1R-bound Ang II, respectively.

The influence of RAAS on rates of change of parathyroid gland and plasma PTH concentrations (denoted $[\text{PTH}]_g$ and $[\text{PTH}]_p$) is incorporated into the corresponding model equations in $\text{Ca}^{2+}$-$\text{Mg}^{2+}$ homeostasis model (equations 1 and 6 in Ref. [28]) via the $F1(\text{ALD, AT1R})$ term:

$$\frac{d[\text{PTH}]_g}{dt} = \frac{k_{\text{prod}}^{\text{PTH}_g}}{1 + \gamma_{\text{prod}}^{1.25(\text{OH})_2\text{D}_3}[1,25(\text{OH})_2\text{D}_3]_p} - k_{\text{deg}}^{\text{PTH}_g}[\text{PTH}]_g \tag{14}$$
$$- F\left([\text{Ca}^{2+}]_p, [\text{Mg}^{2+}]_p\right) \times F1(\text{ALD, AT1R}) \times [\text{PTH}]_g$$

$$\frac{d[\text{PTH}]_p}{dt} = \left(\frac{V_g}{V_p}\right) \times F\left([\text{Ca}^{2+}]_p, [\text{Mg}^{2+}]_p\right) \times F1(\text{ALD, AT1R}) \times [\text{PTH}]_g \tag{15}$$
$$- k_{\text{deg}}^{\text{PTH}_p}[\text{PTH}]_p$$

The effects of AT1R, aldosterone, and plasma $Ca^{2+}$ and $Mg^{2+}$ on PTH are summarized in Fig 1. For descriptions and values of all parameters in the above two equations refer to Ref. [28] and table 2 therein.

## Effect of calcium and magnesium on RSNA

RSNA constricts the afferent arteriole, promotes $Na^+$ and water retention by the kidneys, and elevates blood pressure. These effects are incorporated into the model; see Ref. [33] for the vasoconstrictive effect and see below for the anti-natriuretic effect. The regulation of RSNA is given by

$$rsna = N_{rsna} \times \alpha_{map} \times \alpha_{rap} \times \alpha_{Ca} \times \alpha_{Mg} \tag{16}$$

where $N_{rsna}$ represents the normalized RSNA, and $\alpha_{map}$, $\alpha_{rap}$, $\alpha_{Ca}$, and $\alpha_{Mg}$ represent the effect of mean arterial pressure (MAP), right atrial pressure (RAP), $Ca^{2+}$, and $Mg^{2+}$ on RSNA. All these terms are equal to 1 in the baseline condition. For definitions of $\alpha_{map}$ and $\alpha_{rap}$ refer to Ref. [29] and $\alpha_{Ca}$ and $\alpha_{Mg}$ are defined below.

$Ca^{2+}$ influx through N-type $Ca^{2+}$ channels in sympathetic nerves increases RSNA, which raises the release of nore-pinephrine [26]. $Mg^{2+}$ suppresses RSNA and norepinephrine release by blocking N-type $Ca^{2+}$ channels in sympathetic nerves [27]. Thus, the balance between $Ca^{2+}$ and $Mg^{2+}$ is important for RSNA. We model the regulation of RSNA by $Ca^{2+}$ and $Mg^{2+}$ as

$$\alpha_{Ca} = 0.85 + \frac{0.3}{1 + \frac{[Ca^{2+}]_{p-eq}}{[Ca^{2+}]_p}} \tag{17}$$

$$\alpha_{Mg} = 0.85 + \frac{0.3}{1 + \frac{[Mg^{2+}]_p}{[Mg^{2+}]_{p-eq}}} \tag{18}$$

where $[Ca^{2+}]_{p-eq}$ and $[Mg^{2+}]_{p-eq}$ denote the concentrations of plasma $Ca^{2+}$ and $Mg^{2+}$ at equilibrium, respectively.

## Effect of angiotensin II on bone resorption

Approximately 99% of the total body $Ca^{2+}$ is stored in the bones. Thus, to maintain calcium balance, it is crucial to regu-late the exchange of calcium between the bones and blood. Bone resorption rate is defined as

$$\tau_{res}(PTH, 1,25(OH)_2D_3, AT1R) = \tau_{res}^{min} + \delta_{res}^{max} \times \tag{19}$$
$$\left( f_{PTH}^{res} \times \left( 0.2 + 0.6 \times f_{1,25(OH)_2D_3}^{res} \right) + 0.2 \times f_{AT1R}^{res} \right)$$

where $\tau_{res}^{min}$ and $\delta_{res}^{max}$ denote the minimal and maximal resorption rates, respectively, and their values can be found in Table 2 of Ref. [28]. The Michaelis-Menten terms $f_{PTH}^{res}$ and $f_{1,25(OH)_2D_3}^{res}$ represent the effect of PTH and $1,25(OH)_2D_3$ on resorp-tion, respectively [28]; low $[1,25(OH)_2D_3]_p$ suppresses the PTH-driven resorption, while physiologic or high levels "permit" it. The term $f_{AT1R}^{res}$ represents the effect of AT1R-bound Ang II on bone resorption and is defined below.

Bones express AT1R, and Ang II binds to AT1R to activate osteoclasts through RANKL induction and hence promotes osteoporosis [34]. Treatment with AT1R blocker ameliorates osteoporosis [34]. We assume the following sigmoidal rela-tion between bone resorption and AT1R-bound Ang II.

$$f_{ATIR}^{res} = \frac{[AT1R]^2}{(K_{res}^{AT1R})^2 + [AT1R]^2} \tag{20}$$

where $K_{res}^{AT1R}$ denotes the sensitivity of bone resorption to AT1R-bound Ang II.

## Sodium reabsorption in the proximal tubule and thick ascending limb

The kidneys play a crucial role in long-term blood pressure regulation by maintaining electrolyte and fluid homeostasis via urinary excretion. The proximal tubule and thick ascending limb are the two primary sites along the nephron for Na$^+$ reabsorption. Fractional reabsorption of Na$^+$ along these segments is given by

$$\eta_{\text{pt+tal-sodreab}} = \left(\eta_{\text{pt-sodreab}}^{\text{eq}} \times \gamma_{\text{PTH}} + \eta_{\text{tal-sodrea}}^{\text{eq}} \times \gamma_{\text{CaSR}}\right) \times \gamma_{\text{filsod}} \times \gamma_{\text{AT1R}} \times \gamma_{\text{rsna}} \tag{21}$$

where $\eta_{\text{pt-sodreab}}^{\text{eq}}$ and $\eta_{\text{tal-sodrea}}^{\text{eq}}$ represent fractional Na$^+$ reabsorption in the proximal tubule and thick ascending limb at equilibrium and are assumed to be 0.60 and 0.20, respectively. $\gamma_{\text{PTH}}$, $\gamma_{\text{CaSR}}$, $\gamma_{\text{filsod}}$, $\gamma_{\text{AT1R}}$, and $\gamma_{\text{rsna}}$ denote the effect of PTH, CaSR, filtered Na$^+$ load, AT1R-bound Ang II, and RSNA, respectively, on fractional Na$^+$ reabsorption and are equal to 1 at baseline. The definitions of $\gamma_{\text{filsod}}$, $\gamma_{\text{AT1R}}$, and $\gamma_{\text{rsna}}$ can be found in Ref. [29] and $\gamma_{\text{PTH}}$ and $\gamma_{\text{CaSR}}$ are defined below.

PTH inhibits Na$^+$/H$^+$ exchanger 3 (NHE3) in the proximal tubule and reduces proximal tubule Na$^+$ reabsorption [24]. We model this effect by

$$\gamma_{\text{PTH}} = 0.97 + \frac{0.17}{1 + \left(\frac{[\text{PTH}]_p}{K_{\text{sod-PTH}}}\right)^2} \tag{22}$$

where $K_{\text{sod-PTH}}$ represents the sensitivity of proximal tubule Na$^+$ reabsorption to plasma [PTH].

Ca$^{2+}$ inhibits Na$^+$–K$^+$–Cl$^-$ cotransporter 2 (NKCC2) in the thick ascending limb through the CaSR and hence reduces Na$^+$ reabsorption along this segment [25]. This effect is represented as

$$\gamma_{\text{Ca}} = 0.904 + \frac{0.19}{1 + \left(\frac{[\text{Ca}^{2+}]_p}{K_{\text{sod-Ca}}}\right)^2} \tag{23}$$

where $K_{\text{sod-Ca}}$ represents the sensitivity of Na$^+$ reabsorption in the thick ascending limb to plasma [Ca$^{2+}$].

## Effect of angiotensin II on arterial resistance

In vascular smooth muscle cells, Ang II binds to AT1R and increases intracellular Ca$^{2+}$ both through Ca$^{2+}$ influx from L-type Ca$^{2+}$ channels and Ca$^{2+}$ release from intracellular stores. Administration of angiotensin converting enzyme (ACE) inhibitor to spontaneously hypertensive rats reduced peripheral resistance by $\sim$40% [35] and Ang II infusion to male Wistar rats increased peripheral resistance by $\sim$140% [36]. Based on these observations, we model the effect of AT1R-bound Ang II on arterial resistance as

$$\psi_{\text{AT1R}} = 0.6 + \frac{2.4 - 0.6}{1 + e^{-4.9\left(\frac{[\text{AT1R}]}{[\text{AT1R}]_{\text{eq}}} - 1.187\right)}}. \tag{24}$$

where [AT1R]$_{\text{eq}}$ denotes the concentration of AT1R-bound Ang II at equilibrium.

## Effect of calcium, magnesium, and parathyroid hormone on vascular resistance

Increased Ca$^{2+}$ influx into vascular smooth muscle cells increases vascular resistance. Low and high Ca$^{2+}$ supplementation in male Sprague-Dawley rats decreased and increased vascular resistance by 13.5% and 7%, respectively [20].

Accordingly, we model the effect of plasma $Ca^{2+}$ on vascular resistance as

$$\psi_{Ca} = 0.865 + \frac{0.27}{1 + \frac{[Ca^{2+}]_{p\text{-}eq}}{[Ca^{2+}]_p}} \tag{25}$$

where $[Ca^{2+}]_{p\text{-}eq}$ denotes the concentration of plasma $Ca^{2+}$ at equilibrium.

$Mg^{2+}$ causes vasodilation of vascular smooth muscle cells by blocking $Ca^{2+}$ influx through L-type $Ca^{2+}$ channels. High extracellular $Mg^{2+}$ reduced myogenic tone in wild-type male mice [21]. We model the effect of plasma $Mg^{2+}$ on vascular resistance as

$$\psi_{Mg} = 0.85 + \frac{0.3}{1 + \frac{[Mg^{2+}]_p}{[Mg^{2+}]_{p\text{-}eq}}} \tag{26}$$

where $[Mg^{2+}]_{p\text{-}eq}$ denotes the concentration of plasma $Mg^{2+}$ at equilibrium.

PTH activates PTH receptor 1 on vascular smooth muscle cells, preferentially engaging Gs-coupled G protein → cyclic adenosine monophosphate (cAMP) → protein kinase A (PKA) signaling while weakly stimulating $Ca^{2+}$ mobilization [22]. The net effect is reduced myosin light-chain phosphorylation and smooth muscle relaxation [37]. We modeled the effect of PTH on vascular resistance by assuming that at very low PTH vascular resistance would increase by 20% and at high PTH vascular resistance would decrease by 20%.

$$\psi_{PTH} = 0.8 + \frac{0.4}{1 + \frac{[PTH]_p}{[PTH]_{p\text{-}eq}}} \tag{27}$$

where $[PTH]_{p\text{-}eq}$ denotes the concentration of plasma PTH at equilibrium.

The regulation of arterial resistance is given by

$$R_a = R_{ba} \times \epsilon_{aum} \times \psi_{AT1R} \times \psi_{Ca} \times \psi_{Mg} \times \psi_{PTH} \tag{28}$$

where $R_{ba}$ denotes the basic arterial resistance and $\epsilon_{aum}$ denotes the autonomic multiplier effect and their definitions can be found in Ref. [29]. The parameters $\psi_{AT1R}$, $\psi_{Ca}$, $\psi_{Mg}$, and $\psi_{PTH}$ denote the effect of [AT1R-bound Ang II], $[Ca^{2+}]$, $[Mg^{2+}]$, and [PTH] on vascular resistance and are defined in Eqs (25)–(28).

The regulation of afferent arteriolar resistance is defined as

$$R_{aa} = R_{aa\text{-}eq} \times \beta_{rsna} \times \Sigma_{tgf} \times \Sigma_{myo} \times \psi_{AT1R\text{-}AA} \times \psi_{AT2R\text{-}AA} \times \psi_{Ca} \times \psi_{Mg} \times \psi_{PTH} \tag{29}$$

where $R_{aa\text{-}eq}$ denotes the afferent arteriolar resistance at equilibrium. The parameters $\beta_{rsna}$, $\Sigma_{tgf}$, $\Sigma_{myo}$, $\psi_{AT1R\text{-}AA}$, and $\psi_{AT2R\text{-}AA}$ denote the effect of RSNA, tubuloglomerular feedback (TGF) signal, myogenic response [38], [AT1R-bound Ang II], and [AT2R-bound Ang II] on afferent arteriolar resistance, respectively, and their definitions can be found in Ref. [29]. The parameters $\psi_{Ca}$, $\psi_{Mg}$, and $\psi_{PTH}$ denote the effect of $[Ca^{2+}]$, $[Mg^{2+}]$, and [PTH] on afferent arteriolar resistance and are defined in Eqs (26)–(28).

## Simulating hypertension

Hypertension is a multifactorial disease that may involve a variety of triggers, including overactive RSNA, RAAS, or arterial stiffening. The model parameters adjusted to simulate hypertensive stimuli are the equilibrium values for RSNA, renin

secretion rate, aldosterone secretion rate, and afferent arteriole resistance. We consider five hypertensive cases, featuring primarily an overactive RSNA, increased renin secretion, increased aldosterone secretion, increased vascular tone, or a combination of these stimuli. These models are referred to as HTN-RSNA, HTN-Renin, HTN-ALD, HTN-AA, and HTN-Combined, respectively. For each hypertensive case, parameters were adjusted so that the MAP predicted for the hypertensive model is approximately 120 mmHg. These parameter sets are shown in Table 2.

## Results

### Sensitivity analysis

We performed a local sensitivity analysis by varying each model parameter by $\pm 5\%$ and computing the corresponding steady state. The percentage changes in RSNA, glomerular filtration rate (GFR), MAP, aldosterone concentration, PRA, [PTH], $[1,25(OH)_2D_3]$, $[Mg^{2+}]$, and $[Ca^{2+}]$ corresponding to 5% increase in parameter values are shown in Fig 3. Results obtained by decreasing parameters by 5% exhibit similar trends (not shown).

Changes in parameters that regulate renal $Na^+$ transport have the largest impact on blood pressure-related variables. Specifically, results in Fig 3 indicate that a 5% increase in the fractional reabsorption rates of $Na^+$ in the proximal tubule and thick ascending limb ($\eta^{eq}_{pt+tal\text{-}sodreab}$), distal tubule ($\eta^{eq}_{dt\text{-}sodreab}$), and collecting duct ($\eta^{eq}_{cd\text{-}sodreab}$) causes significant change in aldosterone concentration. Additionally, $\eta^{eq}_{pt+tal\text{-}sodreab}$ has significant impact on GFR (since it determines the macula densa $Na^+$ flow), RSNA, and MAP (by regulating water reabsorption and consequently extracellular fluid volume).

As expected, $Ca^{2+}$-$Mg^{2+}$-related variables are sensitive to changes in (some) parameters that characterize renal $Ca^{2+}$ and $Mg^{2+}$ transport. A 5% increase in minimal thick ascending limb fractional $Mg^{2+}$ reabsorption ($\lambda^0_{Mg\text{-}TAL}$) causes a significant change in plasma [PTH], $[1,25(OH)_2D_3]$, $[Mg^{2+}]$, and $[Ca^{2+}]$. For instance, the increase in $\lambda^0_{Mg\text{-}TAL}$ significantly elevates plasma $[Mg^{2+}]$ and $[Ca^{2+}]$, while suppressing plasma [PTH]. This occurs because increased plasma $[Mg^{2+}]$ directly stimulates $1,25(OH)_2D_3$ production, which in turn enhances intestinal $Ca^{2+}$ absorption, thereby increasing plasma $[Ca^{2+}]$. Consequently, [PTH] decreases due to inhibition from both increased $[1,25(OH)_2D_3]$ and elevated $[Mg^{2+}]$ and $[Ca^{2+}]$. In contrast, a similar change in fractional $Ca^{2+}$ reabsorption in the proximal tubule ($\lambda^0_{Ca\text{-}PT}$) has minimal impact on these variables due to a negative feedback loop: increased plasma $[Ca^{2+}]$ inhibits PTH and $1,25(OH)_2D_3$, dampening further $Ca^{2+}$ changes. Unlike $Ca^{2+}$, the feedback loop of $Mg^{2+}$ with $1,25(OH)_2D_3$ is reinforcing, amplifying the impact of renal $Mg^{2+}$ reabsorption changes.

### Effect of different hypertensive stimuli

As described above, we adjusted selected parameters to investigate the model's responses to different hypertensive stimuli: overactive RSNA, increased renin secretion, increased aldosterone secretion, increased vascular tone, or a combination of these stimuli (Table 2). Also, these parameters were chosen to yield MAP of approximately 120 mmHg, corresponding to an increase of 16%. Model predictions for each of these stimuli are shown in Fig 4.

In the HTN-RSNA case, RSNA induces vasoconstriction and directly stimulates proximal tubule $Na^+$ reabsorption. Taken in isolation, afferent arteriolar constriction would lower GFR. However, the resulting reduced $Na^+$ flow at the macula densa inhibits the TGF signal and causes the afferent arteriole to slightly dilate. The higher MAP would also increase

**Table 2.** Changes in model parameters to simulate five model instances of hypertension, each with different primary trigger(s), labelled RSNA, Renin, ALD, AA, and Combined. RSNA, renal sympathetic nervous activity; ALD, aldosterone; AA, afferent arterial.

|  | *RSNA* | *Renin* | *ALD* | *AA* | *Combined* |
|---|---|---|---|---|---|
| Baseline RSNA | ×1.54 | ×1.0 | ×1.0 | ×1.0 | ×1.28 |
| Renin secretion rate | ×1.0 | ×6.0 | ×1.0 | ×1.0 | ×1.28 |
| ALD secretion rate | ×1.0 | ×1.0 | ×3.0 | ×1.0 | ×1.28 |
| Baseline AA resistance | ×1.0 | ×1.0 | ×1.0 | ×2.9 | ×1.28 |

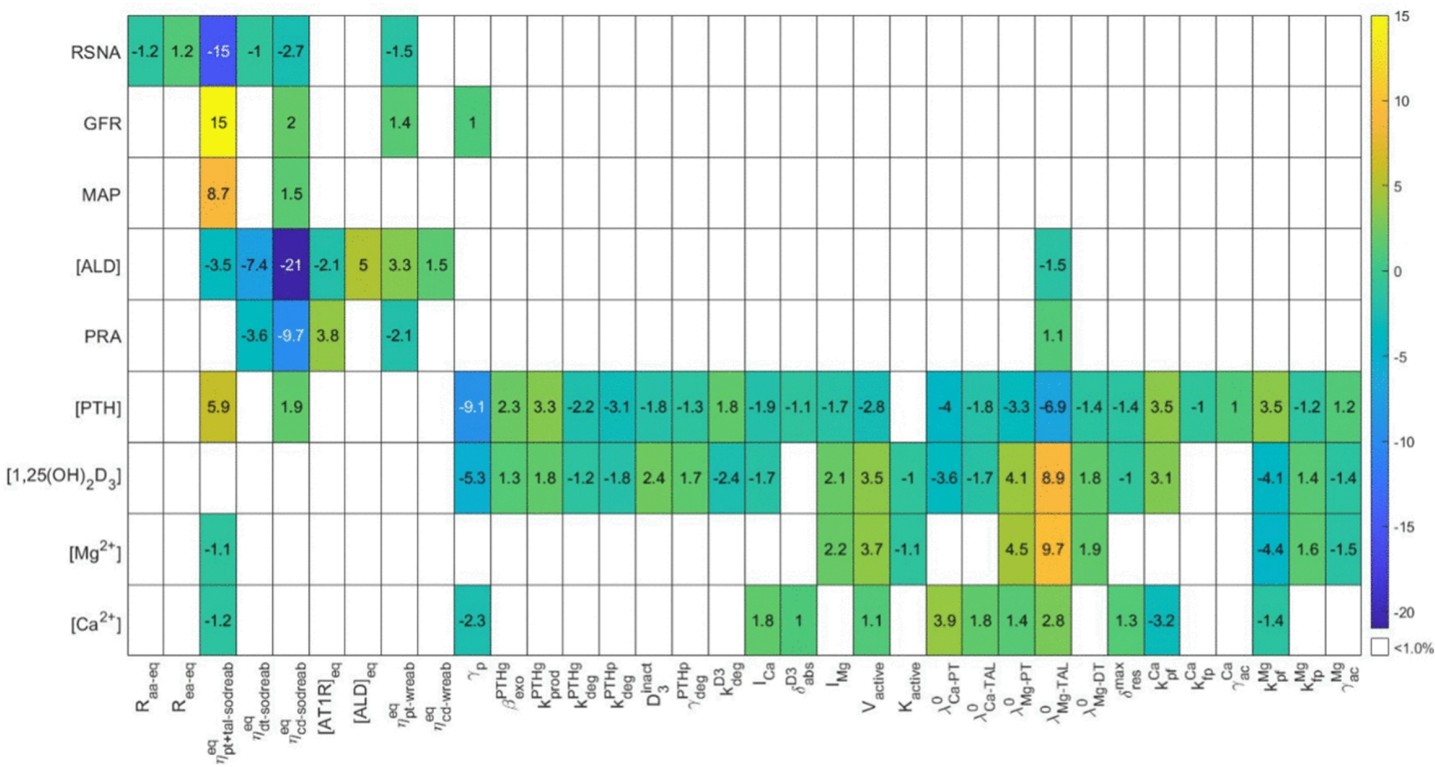

**Fig 3**. **Local sensitivity analysis.** Conducted by increasing individual parameters by 5%. The resulting percent change in model steady state concentrations from baseline is presented here. White indicates the resulting change was less than 1%. RSNA, renal sympathetic nervous activity; GFR, glomerular filtration rate; MAP, mean arterial pressure; [ALD], plasma aldosterone concentration; PRA, plasma renin activity; [PTH], plasma PTH concentration; [1,25(OH)$_2$D$_3$], plasma 1,25(OH)$_2$D$_3$ concentration; [Mg$^{2+}$], plasma Mg$^{2+}$ concentration; [Ca$^{2+}$], plasma Ca$^{2+}$ concentration.

GFR. Together, these factors yield a GFR that is slightly higher than the normotensive values (Fig 4A). RSNA-induced hypertension does not cause any significant changes in plasma Ca$^{2+}$, Mg$^{2+}$, PTH, and 1,25(OH)$_2$D$_3$, levels or on any of Ca$^{2+}$-Mg$^{2+}$ fluxes.

In the HTN-Renin case (hyperreninemia), increased renin secretion elevates the level of AT1R-bound Ang II, which constricts both the afferent and efferent arterioles, but preferentially the latter. As a result, while renal blood flow was predicted to be almost unchanged from baseline, GFR was notably higher (22%; Fig 4A). The significantly increased aldosterone stimulates PTH secretion [40,41], thereby increasing plasma [PTH] by 33%. On one hand, this increased [PTH] inhibits proximal tubular Ca$^{2+}$ and Mg$^{2+}$ reabsorption, while on the other hand it increases Ca$^{2+}$ and Mg$^{2+}$ reabsorption along the thick ascending limb and distal tubule. Now, the majority of Ca$^{2+}$ reabsorption occurs along the proximal tubule, while majority of Mg$^{2+}$ reabsorption occurs along the thick ascending limb. Hence, combined with the higher filtered load, our model predicts a notably higher increase in urinary Ca$^{2+}$ excretion than in urinary Mg$^{2+}$ excretion (Fig 4B). Further, bone resorption increases by 13% under PTH, calcitriol, and AT1R-bound Ang II stimulation. Together, these factors do not cause any significant change in plasma Mg$^{2+}$ and Ca$^{2+}$ concentrations.

The HTN-ALD case simulates primary hyperaldosteronism, where the renin secretion rate remains normal, but there is hypersecretion of aldosterone, with a 30-fold increase above baseline.

Aldosterone secretion rate in primary aldosteronism is difficult to measure. However, we calculated the ratio of plasma aldosterone concentration (PAC) (ng/dL) to plasma renin activity (PRA) (pmol/L/min), which is commonly used in the diagnosis of primary aldosteronism. The proposed cut-off values for PAC/PRA ratio in the literature range from1.6 to 3.1

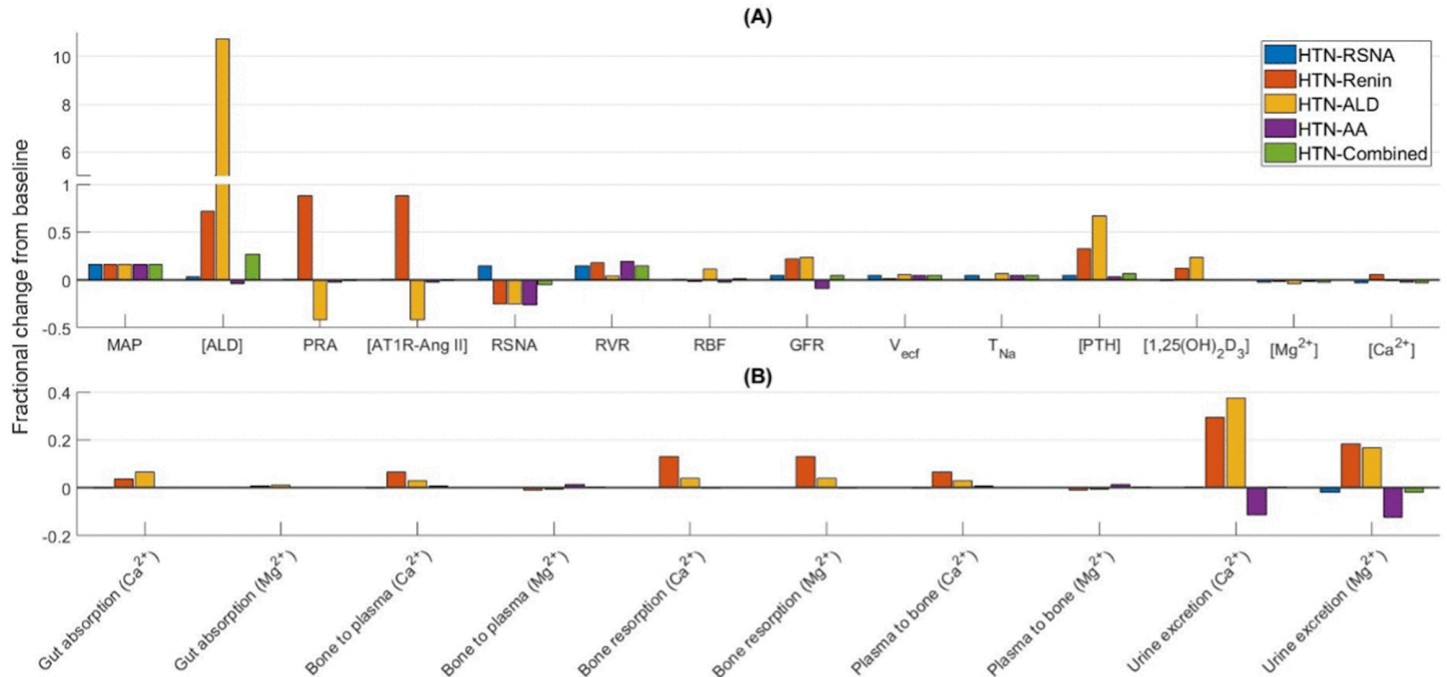

**Fig 4. Effect of hypertensive stimuli.** Fractional change from baseline (denoted by grey line at zero) of model variables (A) and $Ca^{2+}$-$Mg^{2+}$ fluxes (B) under five hypertensive stimuli: overactive RSNA (HTN-RSNA), increased renin secretion (HTN-Renin), increased aldosterone secretion (HTN-ALD), increased vascular tone (HTN-AA), and a combination of these stimuli (HTN-Combined). Bone to plasma and plasma to bone fluxes represent the exchange of $Ca^{2+}$ and $Mg^{2+}$ between plasma and the fast bone pool; bone resorption represents the release of $Ca^{2+}$ and $Mg^{2+}$ from the slow bone pool into plasma. MAP, mean arterial pressure; [ALD], aldosterone concentration; PRA, plasma renin activity; [AT1R-Ang II], plasma AT1R-bound Ang II concentration; RSNA, renal sympathetic nervous activity; RVR; renal vascular resistance; RBF, renal blood flow; GFR, glomerular filtration rate; $V_{ecf}$, extracellular fluid volume; $T_{Na}$, total plasma sodium; [PTH], plasma PTH concentration; $[1,25(OH)_2D_3,]$, plasma $1,25(OH)_2D_3$ concentration; $[Mg^{2+}]$, plasma $Mg^{2+}$ concentration; $[Ca^{2+}]$, plasma $Ca^{2+}$ concentration.

ng/dL per pmol/L/min [42]. We obtained a PAC/PRA ratio of 1.86 ng/dL per pmol/L/min with the predicted plasma aldosterone concentration and PRA values. The high levels of aldosterone cause the kidneys to retain $Na^+$ and increase blood volume, which in turn signals the kidneys to decrease renin production. Our model predicts a 41% decrease in plasma renin activity and AT1R-bound Ang II (Fig 4A). The high aldosterone level also causes a 67% increase in PTH concentration (Fig 4A) [40,41]. This in turn inhibits proximal tubular $Ca^{2+}$ and $Mg^{2+}$ reabsorption, while increasing their reabsorption along the thick ascending limb. Consequently, our model predicts a 37% increase in urinary $Ca^{2+}$ excretion [41,43] while urinary $Mg^{2+}$ excretion is predicted to increase by 17% [44] (Fig 4B). Under the combined effect of increased PTH (67%), increased calcitriol (23%), and decreased AT1R-bound Ang II (41%), bone resorption increases by 3.8% [40,45,46] (Fig 4B). Together, these factors keep plasma $Mg^{2+}$ and $Ca^{2+}$ concentrations near the baseline values [41,44].

Afferent arteriole constriction in the HTN-AA case lowers GFR, whereas the increased MAP increases GFR. Under the influence of these two opposing factors GFR decreases by 8.5%. This in turn reduces proximal tubule $Na^+$ and water reabsorption by a small amount. The increased $Na^+$ delivery to the downstream segments increases $Na^+$ reabsorption in the distal tubule and collecting duct. Hence, total plasma $Na^+$ and extracellular fluid volume increase slightly. In addition, the reduced GFR lowers urinary $Ca^{2+}$ and $Mg^{2+}$ excretions.

The HTN-Combined case involves an overactive RSNA, an overactive RAAS, and increased vascular tone, with the strength of these stimuli chosen so that the predicted MAP is 120 mmHg (Table 2). The high aldosterone level increases [PTH] by 6.4%. However, it does not cause any notable change in any of the $Ca^{2+}$ and $Mg^{2+}$ fluxes.

In summary, the HTN-Renin, HTN-ALD, and HTN-AA cases exhibit significant changes in $Ca^{2+}$ and $Mg^{2+}$ fluxes from baseline. However, plasma $Ca^{2+}$ and $Mg^{2+}$ concentrations are predicted to remain within their respective physiological ranges for all hypertensive stimuli.

**Effect of $Mg^{2+}$, $Ca^{2+}$, and vitamin D deficiency**

Next, we separately simulated 70% dietary $Mg^{2+}$ intake ($I_{Mg}$) restriction and 70% dietary $Ca^{2+}$ intake ($I_{Ca}$) restriction. Vitamin D deficiency was simulated by limiting 25(OH)D (an immediate precursor of $1,25(OH)_2D_3$) to 70%. Each of these restrictions was simulated for 1 month. Model predictions are shown in Fig 5.

Our model predicts a 21% increase in MAP, from 103 mmHg to 125 mmHg, following 1 month of dietary $Mg^{2+}$ restriction (Fig 5A). By contrast, 1 month of dietary $Ca^{2+}$ restriction and 25(OH)D deficiency have comparatively lesser effect on MAP (increases to 119 mmHg and 114 mmHg, respectively). Now, why does $Mg^{2+}$ deficiency have a higher effect on MAP compared to $Ca^{2+}$ and 25(OH)D deficiency? Fig 5B shows the key variables and the interplay between them to answer the above question. Each variable is accompanied by their change over time for each of the three cases.

In the model MAP is calculated as the product of cardiac output and total peripheral resistance; the relative changes in these two factors determine the change in MAP. Dietary $Mg^{2+}$ deficiency markedly lowers plasma $[Mg^{2+}]$, which in turn inhibits PTH secretion. Hence plasma [PTH] drops by 40%. This in turn partially removes the inhibitory effect of PTH on proximal tubule $Na^+$ reabsorption. From Fig 5B, we see that RSNA (maroon bar plot) initially rises under the effect of lower plasma $[Mg^{2+}]$. However, as MAP rises, its inhibitory effect brings down RSNA. Also, GFR (Fig 5B, maroon bar plot) does not change initially, but as MAP rises it increases renal blood flow and hence GFR. Thus, proximal tubule $Na^+$ and water reabsorption is enhanced under the combined effect of increased GFR and removal of the inhibitory effect of PTH. Additionally, the lowered plasma $[Mg^{2+}]$ removes the inhibitory effect on aldosterone secretion causing an increase in plasma aldosterone levels. The high aldosterone levels enhance $Na^+$ and water reabsorption in the distal tubule and collecting duct. Consequently, the extracellular fluid volume increases, which subsequently increases cardiac output by 12%. Now, the increased cardiac output increases the arterial resistance. This is because when more blood flows through any tissue of the body than is required by that tissue for its specific function, the local resistance to blood flow increases progressively to bring the blood flow back towards normal. On the other hand, the lower [PTH] decreases arterial resistance and lower $[Mg^{2+}]$ increases arterial resistance. Under the combined effect of these and cardiac output, arterial resistance increases which in turn increases the total peripheral resistance by 8.3%. Thus, dietary $Mg^{2+}$ deficiency increases cardiac output and total peripheral resistance, which together cause a 21% increase in MAP. In fact, several observational studies, clinical trials, and meta-analyses have shown an inverse relationship between dietary $Mg^{2+}$ intake and hypertension [47–51].

By contrast, dietary $Ca^{2+}$ deficiency significantly increases plasma [PTH] due to the drop in plasma $[Ca^{2+}]$. This reinforces the inhibitory effect of PTH on proximal tubule $Na^+$ reabsorption. In addition, RSNA (Fig 5B, blue bar plot) decreases under the combined effect of lower plasma $[Ca^{2+}]$ and rising MAP. The increasing MAP also increases GFR (Fig 5B, blue bar plot). Furthermore, the lower plasma $[Ca^{2+}]$ removes the inhibitory effect on renin secretion; hence PRA increases, which in turn increases [AT1R-bound Ang II] (Fig 5B, blue bar plot). Together these factors slightly increase proximal tubule $Na^+$ and water reabsorption. Additionally, the increased aldosterone (under the effect of increased [PTH] and [AT1R-bound Ang II]) increases $Na^+$ and water reabsorption in the distal tubule and collecting duct. Hence, the extracellular fluid volume increases, which in turn increases cardiac output. On the other hand, the higher [AT1R-bound Ang II] and [PTH] increase arterial resistance and the lower $[Ca^{2+}]$ decreases arterial resistance. Together these factors increase total peripheral resistance by 15%. This increased resistance inhibits cardiac output. Consequently, cardiac output does not change from the baseline value. Thus, during dietary $Ca^{2+}$ deficiency the 15% increase in MAP is primarily due to increased total peripheral resistance. Several observational studies, clinical trials, and meta-analyses have shown an inverse relationship between dietary $Ca^{2+}$ intake and hypertension [52–54].

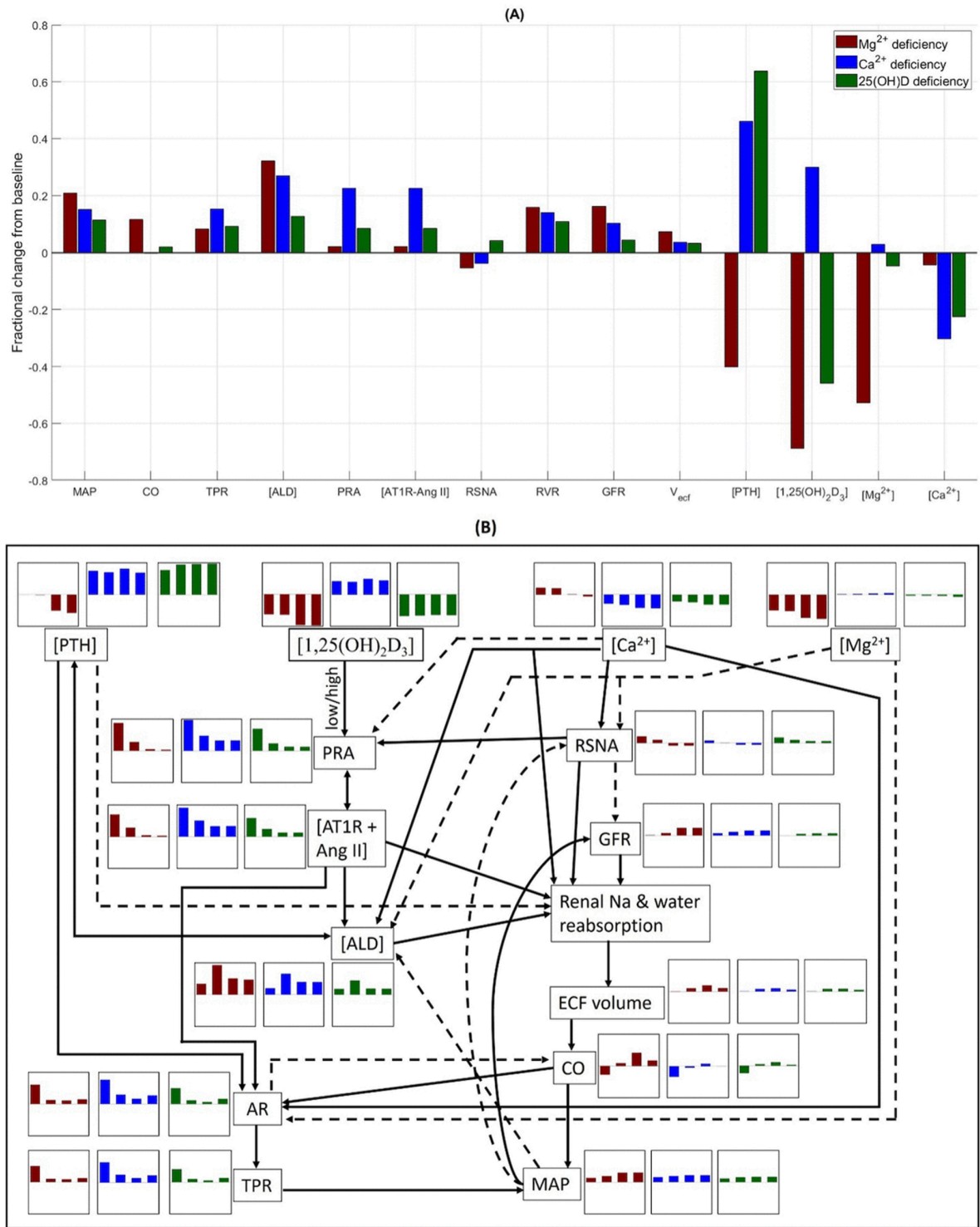

**Fig 5**. **Effect of Mg$^{2+}$, Ca$^{2+}$, and vitamin D$_3$ deficiency.** (A) Fractional change from baseline (denoted by grey line at zero) of model variables under 70% dietary Mg$^{2+}$ intake ($I_{Mg}$) restriction, 70% dietary Ca$^{2+}$ intake ($I_{Ca}$) restriction, and 70% 25(OH)D (precursor of 1,25(OH)$_2$D$_3$) deficiency for 1 month. (B) Interplay between key variables that are affected by dietary Mg$^{2+}$, dietary Ca$^{2+}$, and 25(OH)D deficiency. Each variable is accompanied by their change over time for each of the three cases. Maroon bar plots, dietary Mg$^{2+}$ deficiency; blue bar plots, dietary Ca$^{2+}$ deficiency; green bar plots,

25(OH)D deficiency. The Y-axis range is [−0.7, 0.7]. MAP, mean arterial pressure; CO, cardiac output; TPR, total peripheral resistance; [ALD], aldosterone concentration; PRA, plasma renin activity; [AT1R-Ang II], AT1R-bound Ang II concentration; RSNA, renal sympathetic nervous activity; RVR; renal vascular resistance; GFR, glomerular filtration rate; Vecf, extracellular fluid volume; [PTH], plasma PTH concentration; [1,25(OH)$_2$D$_3$], plasma 1,25(OH)$_2$D$_3$ concentration; [Mg$^{2+}$], plasma Mg$^{2+}$ concentration; [Ca$^{2+}$], plasma Ca$^{2+}$ concentration; AR, arterial resistance.

25(OH)D restriction results in calcitriol (1,25(OH)$_2$D$_3$) deficiency, which lowers intestinal absorption of Ca$^{2+}$ and causes a drop in plasma [Ca$^{2+}$]. This in turn increases PTH secretion. The decreased plasma [1,25(OH)$_2$D$_3$] also partially removes the inhibitory effect on PTH production, resulting in a higher rise in plasma [PTH] compared to the dietary Ca$^{2+}$ deficiency case. In calcitriol deficiency, RAAS, renal Na$^+$ and water reabsorption, and peripheral resistance undergo similar changes as in dietary Ca$^{2+}$ deficiency, though the impact is lower. Consequently, MAP increases by only 11% [55–57].

Thus, our model predicts that the factor that causes Mg$^{2+}$ deficiency to have a higher effect on MAP compared to Ca$^{2+}$ and 25(OH)D deficiency is the increased cardiac output. During Mg$^{2+}$ deficiency, changes in GFR, RSNA, [PTH], [AT1R-bound Ang II], and aldosterone significantly increase renal Na$^+$ and water reabsorption, leading to a considerable increase in extracellular fluid volume and thus cardiac output. In contrast, in Ca$^{2+}$ and 25(OH)D deficiency, changes in these factors cause only modest changes in extracellular fluid volume and hence cardiac output does not change significantly from baseline.

## Primary hyperparathyroidism

Primary hyperparathyroidism was simulated by increasing the baseline PTH synthesis rate ($k_{prod}^{PTHg}$) by factors of 2, 3, 5, 7, and 10. Shown in Fig 6 are the predicted steady-state values of key model variables and fluxes. With these changes, the model predicts plasma [PTH] to increase by 70%, 130%, 237%, 334%, and 468%, respectively. As a result, plasma [1,25(OH)$_2$D$_3$] rises, which in turn greatly enhances intestinal Ca$^{2+}$ absorption, while intestinal Mg$^{2+}$ absorption increases slightly. This is because 1,25(OH)$_2$D$_3$ regulates 45% of intestinal Ca$^{2+}$ absorption but only 12% of intestinal Mg$^{2+}$ absorption. Further, bone resorption increases under the combined stimulation of PTH and 1,25(OH)$_2$D$_3$. Now, PTH inhibits proximal tubular Ca$^{2+}$ and Mg$^{2+}$ reabsorption, where the majority of the Ca$^{2+}$ reabsorption occurs, while stimulating reabsorption along the thick ascending limb and distal tubule, where the majority of the Mg$^{2+}$ reabsorption occurs. Hence, combined with the higher filtered load, our model predicts a notably higher increase in urinary Ca$^{2+}$ excretion than in urinary Mg$^{2+}$ excretion. Together these factors cause plasma [Ca$^{2+}$] to increase by 7% (1.32 mM), 11% (1.37 mM), 15% (1.41 mM), 18% (1.45 mM), and 20% (1.48 mM), respectively, for 2-, 3-, 5-, 7-, and 10-fold increase in $k_{prod}^{PTHg}$. Thus, in all five cases, plasma [Ca$^{2+}$] is above its physiological range (1.1-1.3 mM) indicating hypercalcemia [58]. By contrast, the model predicts no change in plasma [Mg$^{2+}$] [59].

Now let us analyze the effect on RAAS. The elevated plasma [Ca$^{2+}$] inhibits renin secretion, whereas the elevated plasma [1,25(OH)$_2$D$_3$] increases renin secretion. Under their combined effect PRA increases significantly, which in turn increases AT1R-bound Ang II and aldosterone. The increased PTH and Ca$^{2+}$ also increase aldosterone secretion. Thus, plasma aldosterone increases by 21%, 27%, 34%, 37%, and 42%, respectively, for 2-, 3-, 5-, 7-, and 10-fold increase in $k_{prod}^{PTHg}$ [32]. Renal Na$^+$ and water reabsorption increases under the stimulation of AT1R-bound Ang II and aldosterone. An important point to note is that though PTH indirectly enhances Na$^+$ reabsorption through the RAAS, it also directly inhibits proximal tubule Na$^+$ reabsorption. Because of these two opposing factors, extracellular fluid volume and total plasma Na$^+$ increase only slightly. Additionally, the higher [AT1R-bound Ang II] and [Ca$^{2+}$] significantly increase arterial resistance and consequently total peripheral resistance. Thus, the increased extracellular fluid volume increases cardiac output, whereas the increased arterial resistance inhibits cardiac output and together these two factors keep cardiac output close to the baseline value. Consequently, MAP increases by 13% (116 mmHg), 14% (117 mmHg), 15.5% (119 mmHg),

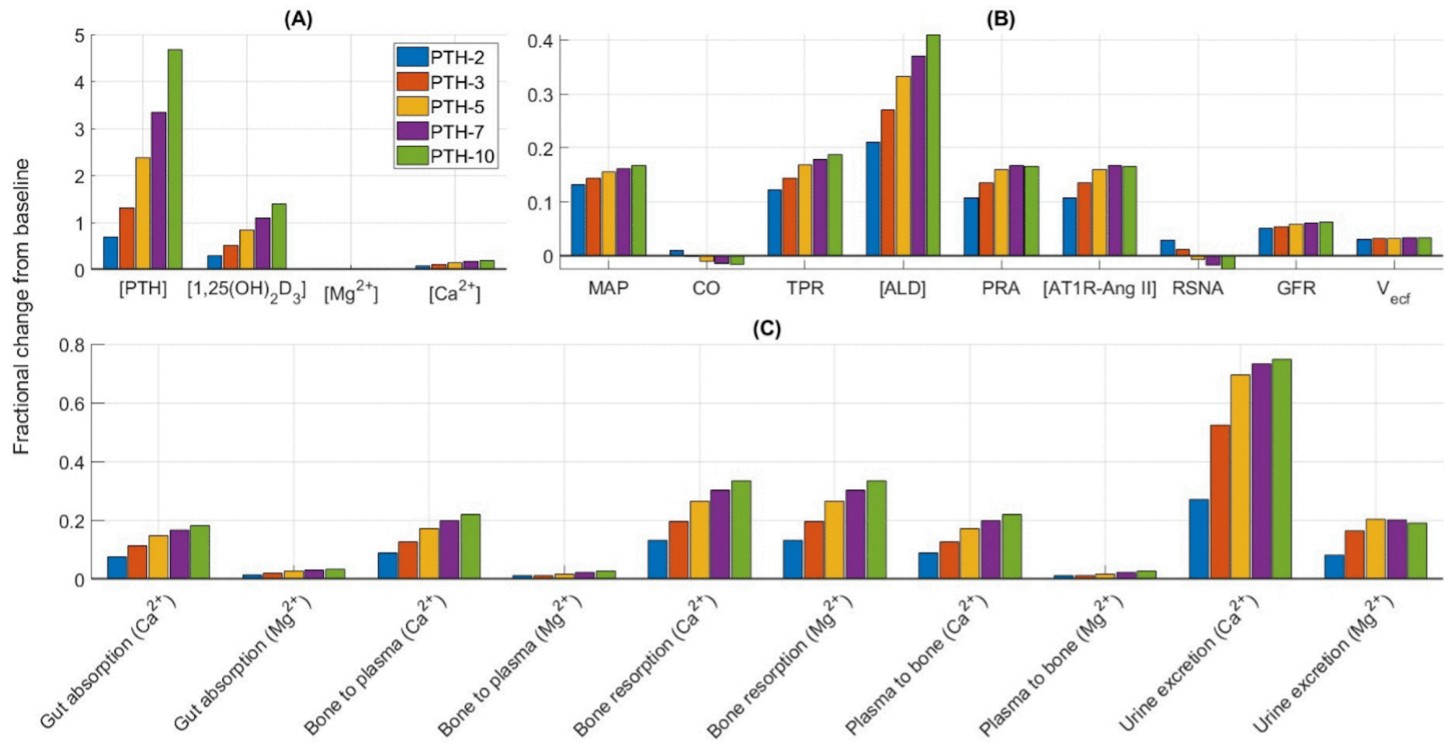

**Fig 6. Simulation results for primary hyperparathyroidism.** Fractional change from baseline (denoted by grey line at zero) of model variables (A, B) and Ca²⁺-Mg²⁺ fluxes (C) after increasing the baseline PTH synthesis rate ($k_{prod}^{PTHg}$) by factors of 2, 3, 5, 7, 10. [PTH], plasma PTH concentration; [1,25(OH)$_2$D$_3$], plasma 1,25(OH)$_2$D$_3$ concentration; [Mg²⁺], plasma Mg²⁺ concentration; [Ca²⁺], plasma Ca²⁺ concentration; MAP, mean arterial pressure; CO, cardiac output; TPR, total peripheral resistance; [ALD], aldosterone concentration; PRA, plasma renin activity; [AT1R-Ang II], AT1R-bound Ang II concentration; RSNA, renal sympathetic nervous activity; GFR, glomerular filtration rate; $V_{ecf}$, extracellular fluid volume.

16% (119.5 mmHg), and 17% (121 mmHg), respectively, for 2-, 3-, 5-, 7-, and 10-fold increase in $k_{prod}^{PTHg}$. Thus, primary hyperparathyroidism causes hyperaldosteronism [32,60] and increases the risk of hypertension. In fact primary hyperparathyroidism has been associated with increased risk of hypertension, with a prevalence ranging from 40-65% and parathyroidectomy has resulted in substantial fall in both mean systolic and diastolic blood pressures [61–65]

Osteopenia and osteoporosis are known to be frequent complications of primary hyperparathyroidism [66–68]. Indeed our model predicts significant increase in bone resorption due to increased PTH and AT1R-bound Ang II.

## Discussion

We have developed an integrated model of blood pressure regulation and Ca²⁺-Mg²⁺ homeostasis to gain insights into the complex interactions that govern these physiological processes. Our study highlights the bidirectional relationship between these two systems, where alterations in one can lead to significant changes in the other. Specifically, we demonstrated how dysregulation in Ca²⁺ and Mg²⁺ levels can affect blood pressure control, and how dysregulation in blood pressure control, such as in hypertension, can impact Ca²⁺ and Mg²⁺ homeostasis.

Hypertension has been associated with elevated plasma PTH [69], increased Ca²⁺ excretion [69,70], and reduced bone mineral density [71]. We simulated hypertension using different stimuli, namely, increased RSNA, increased renin secretion, increased aldosterone secretion, increased vascular tone, and a combination of all these stimuli. Simulation

results (Fig 4) suggest that hypertensive states driven by elevated renin (HTN-Renin) and aldosterone (HTN-ALD) secretions significantly influence PTH, calcitriol, and the renal handling of $Ca^{2+}$ and $Mg^{2+}$. Notably, both conditions exhibit increased urinary excretion of $Ca^{2+}$ and $Mg^{2+}$, as well as elevated bone resorption (Fig 4). Primary hyperaldosteronism, which results from increased aldosterone secretion, has been associated with hyperparathyroidism [40,41] and increased urinary excretions of $Ca^{2+}$ [41,43] and $Mg^{2+}$ [44], although plasma $Ca^{2+}$ and $Mg^{2+}$ remain normal [41,44]. In fact, primary aldosteronism has been associated with nephrocalcinosis [72] and nephrolithiasis [73] caused by hypercalciuria. In addition, primary aldosteronism also causes bone loss and reduced bone mineral density [40,45,46]. Now, hyperreninemia, caused by increased renin secretion, has been associated with hyperaldosteronism [74,75]. Thus, the elevated aldosterone levels in hyperreninemia also increase PTH levels and hence urinary excretions of $Ca^{2+}$ and $Mg^{2+}$. Our model predicts a significantly higher increase in bone resorption rate in the HTN-Renin case compared to the HTN-ALD case (Fig 4). This is because increased renin secretion increases ATIR-bound Ang II, which increases the risk of osteoporosis. In fact, the use of angiotensin-converting enzyme (ACE) inhibitors have been associated with higher bone mineral density and reduced risk of fracture [76,77]. Despite these changes in urinary excretion and bone resorption rate, the model simulations indicate that plasma $Ca^{2+}$ and $Mg^{2+}$ concentrations remain relatively stable across all simulated hypertensive conditions (HTN-RSNA, HTN-Renin, HTN-ALD, HTN-AA, HTN-Combined) (Fig 4), suggesting that the body's compensatory mechanisms effectively maintain mineral homeostasis even under pathological conditions.

Cross-sectional and longitudinal epidemiological studies have consistently reported an inverse relationship between dietary $Mg^{2+}$ and blood pressure and/or hypertension [78–83]. Additionally, two large meta-analyses of randomized trials reported that $Mg^{2+}$ supplementation significantly lowers blood pressure [49,50]. Hypomagnesemia has also been associated with pre-eclampsia [84–86], a pregnancy specific hypertensive disorder, and $Mg^{2+}$ supplementation has been reported to reduce the risk of eclampsia in pregnant women by over 50% [87]. Together, these studies highlight the importance of $Mg^{2+}$ homeostasis in blood pressure regulation. We conducted simulations to investigate the mechanisms responsible for increased mean arterial pressure during $Mg^{2+}$ deficiency. Model simulations indicated that $Mg^{2+}$ deficiency increased both cardiac output and total peripheral resistance (Fig 4). The increase in cardiac output was primarily due to decreased PTH and increased aldosterone, which increased renal $Na^+$ and water reabsorption. Total peripheral resistance increased mainly due to increased cardiac output and removal of the inhibitory effect of $Mg^{2+}$.

Several epidemiological studies have reported an inverse relationship between dietary $Ca^{2+}$ intake and blood pressure and/or hypertension [52,53,78,79,82,85,88]. Simulation results are aligned with the observations from these studies. Unlike $Mg^{2+}$, $Ca^{2+}$ deficiency was predicted to increase the total peripheral resistance but did not change cardiac output (Fig 5). Additionally, our model predicted that dietary $Mg^{2+}$ deficiency has a stronger effect on mean arterial pressure than dietary $Ca^{2+}$ deficiency (Fig 5). In fact, two cross-sectional studies have reported dietary $Mg^{2+}$ intake to have a stronger association with blood pressure compared to dietary $Ca^{2+}$ intake [82,89]. Since calcitriol deficiency significantly lowers plasma $[Ca^{2+}]$, it has same effect as dietary $Ca^{2+}$ deficiency on mean arterial pressure, i.e., an inverse association [55–57,90,91], although the impact is slightly lower.

Primary hyperparathyroidism has been associated with osteoporosis [66–68] and hypertension [61–65]. Our model predictions suggested that the elevated plasma PTH, $Ca^{2+}$, and calcitriol are primarily responsible for these two disorders (Fig 5). These three factors overactivate the RAAS and increase vascular resistance, which in turn increases the mean arterial pressure. The elevated PTH, calcitriol, and AT1R-bound Ang II also promote bone loss (Fig 5).

In summary, we have developed a computational model representing the interplay between $Ca^{2+}$ and $Mg^{2+}$ homeostasis and blood pressure regulation in a male rat. The model was used to understand the underlying mechanisms involved in (i) regulating $Ca^{2+}$ and $Mg^{2+}$ balance during different hypertensive stimuli, (ii) blood pressure regulation during dietary $Mg^{2+}$ deficiency, dietary $Ca^{2+}$ deficiency, and vitamin D deficiency, and (iii) $Ca^{2+}$, $Mg^{2+}$, and blood pressure regulation during primary hyperparathyroidism.

## Limitations of the study

The present model is based primarily on a male rat. However, there are many known sex differences in blood pressure regulation [29,33,92], and in the renal handling of $Na^+$, $Ca^{2+}$, and $Mg^{2+}$ [93–96]. A worthwhile extension would be to develop sex-specific models for $Ca^{2+}$, $Mg^{2+}$, and blood pressure regulation under various physiological and pathophysiological conditions. Furthermore, there are significant interspecies differences between rat and human in terms of $Ca^{2+}$, $Mg^{2+}$, and blood pressure regulation, including intestinal $Ca^{2+}$ and $Mg^{2+}$ absorption, bone remodeling, and baseline hemodynamics. As such, the translation from rat to human is not straightforward. While qualitative feedback structure and directionality of hormonal effects may translate well, other more quantitative predictions such as time constants and gain values may not.

The actions of the transporters and channels along the nephron cell membranes that regulate $Na^+$, $Ca^{2+}$, $Mg^{2+}$, and fluid balance are represented implicitly in our model. A possible extension of the model would be to explicitly model these transporters and channels as done in epithelial transport models [94,97,98]). The benefit of coupling individual nephron with whole kidney dynamics would be in simulating the administration of drugs that target these transporters. The exact action of the drug could then be simulated instead of inferred.

Dysregulated PTH and $Ca^{2+}$ in hyper- and hypoparathyroidism exert important cardiovascular effects, including altered contractility, electrical instability, and ventricular remodeling. While our model captures their systemic interactions with mineral electrolyte and blood pressure regulation, their direct myocardial and electrophysiological actions are not explicitly included.

The present model does not include potassium ($K^+$), which is a key electrolyte in blood pressure regulation. Higher $K^+$ levels have been associated with lower blood pressure [99–101]. Its blood pressure-lowering effects stem from its capacity to induce vasodilation, a process mediated by vascular cell hyperpolarization [102]. Additionally, $K^+$ influences blood pressure by increasing $Na^+$ excretion, modulating baroreceptor sensitivity, reducing vasoconstrictive sensitivity to norepinephrine and angiotensin II, and increasing Na-K-ATPase activity [102]. A future extension of the model would be inclusion of $K^+$ regulation [103,104] and its interaction with various components of blood pressure regulation.

## Conclusion

This study introduces a comprehensive computational framework that unifies blood pressure regulation with $Ca^{2+}$–$Mg^{2+}$ homeostasis, enabling systematic exploration of how hormonal, renal, and vascular mechanisms jointly maintain cardiovascular and electrolyte stability. Through simulations of hypertension driven by distinct physiological triggers, mineral and vitamin D deficiencies, and primary hyperparathyroidism, the model reveals how perturbations in one system propagate through the other. The results highlight magnesium as a particularly influential regulator of blood pressure, show how RAAS activation can reshape calcium and magnesium fluxes without markedly altering plasma concentrations, and clarify how elevated PTH, calcitriol, and Ang II contribute to both hypertension and bone loss in hyperparathyroidism.

Overall, the model captures the multilayered feedbacks that couple mineral metabolism to cardiovascular control and provides mechanistic insights consistent with diverse experimental and clinical observations. By identifying pathways most sensitive to perturbation, this work offers a platform for hypothesis generation and for evaluating how dietary interventions, endocrine disorders, and therapeutic strategies might influence blood pressure and mineral balance. Future extensions incorporating sex differences, nephron-scale transporter modeling, and potassium regulation will further enhance the model's predictive and translational potential.

## Acknowledgments

The authors acknowledge the support of the University of Waterloo.

## Author contributions

**Conceptualization:** Pritha Dutta, Anita T. Layton.

**Data curation:** Pritha Dutta, Anita T. Layton.

**Formal analysis:** Pritha Dutta, Anita T. Layton.

**Funding acquisition:** Anita T. Layton.

**Investigation:** Pritha Dutta, Anita T. Layton.

**Methodology:** Pritha Dutta, Anita T. Layton.

**Project administration:** Anita T. Layton.

**Resources:** Anita T. Layton.

**Supervision:** Anita T. Layton.

**Validation:** Pritha Dutta, Anita T. Layton.

**Visualization:** Pritha Dutta.

**Writing – original draft:** Pritha Dutta.

**Writing – review & editing:** Pritha Dutta, Anita T. Layton.

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
