## [Decision Letter · Decision Letter 0]

22 Dec 2025

PONE-D-25-64284Unraveling the Mechanistic Links Between Blood Pressure Regulation and Calcium-Magnesium Homeostasis: Insights into Hypertension, Hyperparathyroidism, and Mineral DisordersPLOS One

Dear Dr. Layton,

Thank you for submitting your manuscript to PLOS ONE. After careful consideration, we feel that it has merit but does not fully meet PLOS ONE’s publication criteria as it currently stands. Therefore, we invite you to submit a revised version of the manuscript that addresses the points raised during the review process.

We look forward to receiving your revised manuscript.

Kind regards,

Toshio Matsumoto

Academic Editor

PLOS One

Journal Requirements:

2. Please note that PLOS One has specific guidelines on code sharing for submissions in which author-generated code underpins the findings in the manuscript. In these cases, all author-generated code must be made available without restrictions upon publication of the work. Please review our guidelines at https://journals.plos.org/plosone/s/materials-and-software-sharing#loc-sharing-code and ensure that your code is shared in a way that follows best practice and facilitates reproducibility and reuse.

This research was supported in part by the Natural Sciences and Engineering Research Council of Canada, via a Discovery award RGPIN-2025-03958 to AT Layton.

This research was supported in part by the Natural Sciences and Engineering Research Council of Canada, via a Discovery award RGPIN-2025-03958 to AT Layton.

This research was supported in part by the Natural Sciences and Engineering Research Council of Canada, via a Discovery award RGPIN-2025-03958 to AT Layton.

This research was supported in part by the Natural Sciences and Engineering Research Council of Canada, via a Discovery award RGPIN-2025-03958 to AT Layton.

Additional Editor Comments:

As indicated in the comments from the two expert reviewers, there are problems in the assumptions of the proposed model. These include but are not limited to the following:

1. This computational model is primarily based upon the data in male rats. The authors should acknowledge the limitations for extrapolating the results in male rats to humans.

2. It is unclear what other factors than Ca, Mg and calciotropic hormones, such as SNS tone, etc., are incorporated in the calculation model.

3. Descriptions about native vitamin D, D3, 25OHD and 1,25(OH)2D should be corrected. In addition, physiological concentrations of 1,25(OH)2D do not enhance bone resorption.

4. PTH and PTHrP act via PTH1R on the vascular wall to enhance vasodilation. How these facts reconcile with the authors’ assumption that high levels of PTH enhance vasoconstriction?

5. [Ca2+]p-eq and [Mg2+]p-eq should be more clearly defined.

If the authors can address all these concerns and criticisms, the revised manuscript can be reconsidered for publication in PLOS ONE.

Reviewers' comments:

Reviewer's Responses to Questions

**Comments to the Author**

1. Is the manuscript technically sound, and do the data support the conclusions?

Reviewer #1: Partly

Reviewer #2: Partly

2. Has the statistical analysis been performed appropriately and rigorously?

Reviewer #1: N/A

Reviewer #2: Yes

3. Have the authors made all data underlying the findings in their manuscript fully available?

Reviewer #1: Yes

Reviewer #2: Yes

4. Is the manuscript presented in an intelligible fashion and written in standard English?

Reviewer #1: Yes

Reviewer #2: Yes

5. Review Comments to the Author

Reviewer #1: General comments

In this manuscript, the authors developed a computational model that integrates blood pressure, Ca²⁺/Mg²⁺ levels, and calciotropic hormones such as PTH and vitamin D. This model could enhance our understanding of certain pathophysiological conditions, such as cardiovascular changes in primary hyperparathyroidism. However, the model appears to rely on numerous assumptions regarding unestablished interactions between relevant factors, which raises some doubts about its validity.

Specific comments

1) In Figure 2, MAP is determined by cardiac output and peripheral vascular resistance, which is fine. But cardiac output seems to be a function of venous return/circulating volume here. Various factors including SNS tone, PTH, Ca and others affect cardiac rhythm and contractile function of the heart. I wonder how the current computational model could incorporate the factor of cardiac function.

2) Page 7: 1,25D does not stimulate bone resorption at least in near-physiological concentrations. What is the rationale for the assumption that both PTH and 1,25D affect bone resorption?

3) Page 8: The authors describe that they modeled the effect of PTH on vascular resistance by assuming that at very low PTH vascular resistance would decrease by 20% and at high PTH vascular resistance would increase by 20%. However, PTH is a well-known vasodilator. I wonder how this assumption could be justified.

Reviewer #2: The authors developed a computational model to investigate the interaction between blood pressure control and regulation of calcium and magnesium metabolism. They found that there are multiple interaction pathways between these. They also simulated a couple of diseases to further delineate these interactions.

Major

1. Because this model is based primarily on a male rat, it is possible that all the results cannot be applied to human. For example, calcium deficiency does not cause 30% decrease in serum calcium in humans. Still, the authors indicate that the model provides interactions that are consistent with diverse clinical observations. It should be mentioned which results can be specific for rats.

2. While PTH is shown to increase arterial resistance, several reports indicate that PTH is a vasodilator. How do you explain this?

3. Calcium and magnesium are shown to have opposite effects on RSBA and vascular resistance (p. 6, 8). It is not clear how effects of these ions can be calculated in the similar formulas (26 and 27). Maybe, [Ca2+]p-eq and [Mg2+]p-eq need to be more clearly explained.

Minor

1. Vitamin D and vitamin D3 are not the same. 25(OH)D is not an inactive form of vitamin D3, but 25-hydroxylated product of vitamin D. Wordings concerning vitamin D should be confirmed.

2. Henle’s loop is a component of distal tubule (Figure 1).

3. It is not clear what right arterial pressure indicates (Figure 2).

4. References should be presented in a fixed format.

6. PLOS authors have the option to publish the peer review history of their article (what does this mean?). If published, this will include your full peer review and any attached files.

Reviewer #1: No

Reviewer #2: No

---

## [Author Response · Author response to Decision Letter 1]

29 Dec 2025

Additional Editor Comments:

1. This computational model is primarily based upon the data in male rats. The authors should acknowledge the limitations for extrapolating the results in male rats to humans.

Reply: Yes, there are significant interspecies differences between rat and human in terms of Ca2+, Mg2+, and blood pressure regulation, including intestinal Ca2+ and Mg2+ absorption, bone remodeling, and baseline hemodynamics. As such, the translation from rat to human is not straightforward. While qualitative feedback structure and directionality of hormonal effects may translate well, other more quantitative predictions such as time constants and gain values may not. These limitations are now acknowledged in the Discussion.

2. It is unclear what other factors than Ca, Mg and calciotropic hormones, such as SNS tone, etc., are incorporated in the calculation model.

Reply: The blood pressure regulation model extends previously published frameworks by Leete et al., which themselves build on foundational models developed by Hallow et al., Karaaslan et al., and Guyton et al. These models describe the coupled interactions among the cardiovascular system, renal function, renal sympathetic nervous activity, and the renin–angiotensin endocrine axis. Because these components have been described in detail elsewhere, we do not reproduce all model equations or provide a complete description of every subsystem. Instead, we focus on those components that are directly regulated by calcium, magnesium, and calcitropic hormones. Additional text and references have been included in the Methods section to clarify how other physiological factors are represented within the model.

3. Descriptions about native vitamin D, D3, 25OHD and 1,25(OH)2D should be corrected. In addition, physiological concentrations of 1,25(OH)2D do not enhance bone resorption.

Reply: The usage of vitamin D, D3, 25(OH)D and 1,25(OH)2D3 has been corrected.

Yes, physiological concentrations of 1,25(OH)2D3 do not independently stimulate bone resorption but instead act permissively to support PTH-mediated resorption. We have revised Eq. (20) to better reflect this relationship and added a clarifying sentence immediately below the equation.

4. PTH and PTHrP act via PTH1R on the vascular wall to enhance vasodilation. How these facts reconcile with the authors’ assumption that high levels of PTH enhance vasoconstriction?

Reply: Yes, you are correct that although PTH can elicit modest Ca²⁺ entry signals in vascular smooth muscle cells, the net integrated signaling downstream of PTH1R favors smooth muscle relaxation rather than contraction. Accordingly, we revised the formulation describing the effect of PTH on vascular resistance (Eq. 28) such that vascular resistance increases by up to 20% at very low PTH levels and decreases by up to 20% at high PTH levels. Despite this vasodilatory influence, vascular resistance remains elevated in hyperparathyroidism because the vasoconstrictive effect of AT1R-bound angiotensin II predominates.

5. [Ca2+]p-eq and [Mg2+]p-eq should be more clearly defined.

Reply: My mistake. The numerator and denominator of the [Mg2p]p/[Mg2p]p-eq fraction were reversed. This has now been corrected and the formulae should make more sense.

Reviewer #1:

1) In Figure 2, MAP is determined by cardiac output and peripheral vascular resistance, which is fine. But cardiac output seems to be a function of venous return/circulating volume here. Various factors including SNS tone, PTH, Ca and others affect cardiac rhythm and contractile function of the heart. I wonder how the current computational model could incorporate the factor of cardiac function.

Reply: The blood pressure regulation model is an expansion of published models by Leete et al., which is in turn based on the models by Hallow et al., Karaaslan et al., and Guyton et al. These models describe the interactions among the cardiovascular system, the renal system, the renal sympathetic nervous system, and the endocrine (renin-angiotensin) system. Since these models have been published, we did not reproduce all the model equations or describe every component, but only those that are directly regulated by Ca, Mg, and calcitropic hormones. We have added text in the Methods section and included references to make clear that other factors are represented in the model.

2) Page 7: 1,25D does not stimulate bone resorption at least in near-physiological concentrations. What is the rationale for the assumption that both PTH and 1,25D affect bone resorption?

Reply: Yes, you are correct that physiological concentrations of 1,25(OH)2D3 does not enhance bone resorption; they merely “permit” PTH-driven resorption. We have corrected Eq. (20) to better represent this effect, and added a sentence below that equation to clarify.

3) Page 8: The authors describe that they modeled the effect of PTH on vascular resistance by assuming that at very low PTH vascular resistance would decrease by 20% and at high PTH vascular resistance would increase by 20%. However, PTH is a well-known vasodilator. I wonder how this assumption could be justified.

Reply: Yes, you are correct that although PTH can increase Ca²⁺ entry signals in vascular smooth muscle cells, the net integrated signaling downstream of PTH1R strongly favors relaxation, not contraction. We have modified the term that describes the effect of PTH on vascular resistance (Eq. 28), such that at very low PTH vascular resistance would increase by 20% and at high PTH vascular resistance would decrease by 20%. Vascular resistance still increases in hyperparathyroidism, because the vasoconstrictive effect of AT1R-bound Ang II dominates.

Reviewer #2:

Major

1. Because this model is based primarily on a male rat, it is possible that all the results cannot be applied to human. For example, calcium deficiency does not cause 30% decrease in serum calcium in humans. Still, the authors indicate that the model provides interactions that are consistent with diverse clinical observations. It should be mentioned which results can be specific for rats.

Reply: The reviewer is correct that there are significant interspecies differences between rat and human in terms of Ca2+, Mg2+, and blood pressure regulation, including intestinal Ca2+ and Mg2+ absorption, bone remodeling, and baseline hemodynamics. As such, the translation from rat to human is not straightforward. While qualitative feedback structure and directionality of hormonal effects may translate well, other more quantitative predictions such as time constants and gain values may not. These limitations are now acknowledged in the Discussion.

2. While PTH is shown to increase arterial resistance, several reports indicate that PTH is a vasodilator. How do you explain this?

Reply: Yes, you are correct that although PTH can increase Ca²⁺ entry signals in vascular smooth muscle cells, the net integrated signaling downstream of PTH1R strongly favors relaxation, not contraction. We have modified the term that describes the effect of PTH on vascular resistance (Eq. 28), such that at very low PTH vascular resistance would increase by 20% and at high PTH vascular resistance would decrease by 20%. Vascular resistance still increases in hyperparathyroidism, because the vasoconstrictive effect of AT1R-bound Ang II dominates.

3. Calcium and magnesium are shown to have opposite effects on RSBA and vascular resistance (p. 6, 8). It is not clear how effects of these ions can be calculated in the similar formulas (26 and 27). Maybe, [Ca2+]p-eq and [Mg2+]p-eq need to be more clearly explained.

Reply: My mistake. The numerator and denominator of the [Mg2p]p/[Mg2p]p-eq fraction were reversed. This has now been corrected and the formulae should make more sense.

Minor

1. Vitamin D and vitamin D3 are not the same. 25(OH)D is not an inactive form of vitamin D3, but 25-hydroxylated product of vitamin D. Wordings concerning vitamin D should be confirmed.

Reply. The usage of vitamin D, D3, 25(OH)D and 1,25(OH)2D3 has been corrected.

2. Henle’s loop is a component of distal tubule (Figure 1).

Reply: In Fig. 2(A), the “proximal tubule” includes both the proximal tubule and the loop of Henle, to better correspond to Eq. 22. For clarify this unconventional representation is now noted in the caption.

3. It is not clear what right arterial pressure indicates (Figure 2).

Reply: That was a typo, our apologies. It should be right atrial pressure. The figure has been corrected.

4. References should be presented in a fixed format.

Reply: We have made the bibliography format consistent

---

## [Decision Letter · Decision Letter 1]

8 Jan 2026

PONE-D-25-64284R1Unraveling the Mechanistic Links Between Blood Pressure Regulation and Calcium-Magnesium Homeostasis: Insights into Hypertension, Hyperparathyroidism, and Mineral DisordersPLOS One

Dear Dr. Layton,

Thank you for submitting your manuscript to PLOS ONE. After careful consideration, we feel that it has merit but does not fully meet PLOS ONE’s publication criteria as it currently stands. Therefore, we invite you to submit a revised version of the manuscript that addresses the points raised during the review process.

We look forward to receiving your revised manuscript.

Kind regards,

Toshio Matsumoto

Academic Editor

PLOS One

**Journal Requirements:**

Reviewers' comments:

Reviewer's Responses to Questions

**Comments to the Author**

1. If the authors have adequately addressed your comments raised in a previous round of review and you feel that this manuscript is now acceptable for publication, you may indicate that here to bypass the “Comments to the Author” section, enter your conflict of interest statement in the “Confidential to Editor” section, and submit your "Accept" recommendation.

Reviewer #1: (No Response)

Reviewer #2: All comments have been addressed

2. Is the manuscript technically sound, and do the data support the conclusions?

Reviewer #1: Partly

Reviewer #2: Yes

3. Has the statistical analysis been performed appropriately and rigorously?

Reviewer #1: N/A

Reviewer #2: (No Response)

4. Have the authors made all data underlying the findings in their manuscript fully available?

Reviewer #1: No

Reviewer #2: (No Response)

5. Is the manuscript presented in an intelligible fashion and written in standard English?

Reviewer #1: Yes

Reviewer #2: (No Response)

6. Review Comments to the Author

Reviewer #1: The manuscript has been substantially improved following the revision. However, a significant concern remains regarding the previous comment #1. The authors state that they expanded the previously published model and focused only on factors that are directly regulated by Ca, Mg, and calciotropic hormones. It is well established that changes in PTH and Ca significantly affect the cardiovascular system in both hyperparathyroidism and hypoparathyroidism, at least in part due to their direct effects on cardiac muscle. The authors should recognize that the lack of consideration for the effects of calciotropic hormones and minerals on cardiac muscle in the current model represents a substantial limitation. This point should be clearly discussed in the text.

Reviewer #2: The authors appropriately revised the manuscript according to the reviewers' comments. I have no other comments.

7. PLOS authors have the option to publish the peer review history of their article (what does this mean?). If published, this will include your full peer review and any attached files.

Reviewer #1: No

Reviewer #2: No

---

## [Author Response · Author response to Decision Letter 2]

9 Jan 2026

Reviewer #1:

The manuscript has been substantially improved following the revision. However, a significant concern remains regarding the previous comment #1. The authors state that they expanded the previously published model and focused only on factors that are directly regulated by Ca, Mg, and calciotropic hormones. It is well established that changes in PTH and Ca significantly affect the cardiovascular system in both hyperparathyroidism and hypoparathyroidism, at least in part due to their direct effects on cardiac muscle. The authors should recognize that the lack of consideration for the effects of calciotropic hormones and minerals on cardiac muscle in the current model represents a substantial limitation. This point should be clearly discussed in the text.

Reply: The reviewer is correct that in both hyper- and hypoparathyroidism, altered PTH and Ca²⁺ levels are known to profoundly affect cardiovascular function, contributing to hypertension, arrhythmias, myocardial hypertrophy, and heart failure. These effects arise not only through systemic changes in vascular tone and renal sodium handling, but also through direct actions of PTH and Ca²⁺ on cardiac muscle, which are not explicitly represented in the present model. While our model captures their systemic interactions with mineral electrolyte and blood pressure regulation, their direct myocardial and electrophysiological actions are not explicitly included. This limitation is now discussed (under Limitations of the study).

---

## [Decision Letter · Decision Letter 2]

13 Jan 2026

Unraveling the Mechanistic Links Between Blood Pressure Regulation and Calcium-Magnesium Homeostasis: Insights into Hypertension, Hyperparathyroidism, and Mineral Disorders

PONE-D-25-64284R2

Dear Dr. Layton,

We’re pleased to inform you that your manuscript has been judged scientifically suitable for publication and will be formally accepted for publication once it meets all outstanding technical requirements.

Kind regards,

Toshio Matsumoto

Academic Editor

PLOS One

**Comments to the Author**

1. If the authors have adequately addressed your comments raised in a previous round of review and you feel that this manuscript is now acceptable for publication, you may indicate that here to bypass the “Comments to the Author” section, enter your conflict of interest statement in the “Confidential to Editor” section, and submit your "Accept" recommendation.

Reviewer #1: All comments have been addressed

2. Is the manuscript technically sound, and do the data support the conclusions?

Reviewer #1: Yes

3. Has the statistical analysis been performed appropriately and rigorously?

Reviewer #1: N/A

4. Have the authors made all data underlying the findings in their manuscript fully available?

Reviewer #1: No

5. Is the manuscript presented in an intelligible fashion and written in standard English?

Reviewer #1: Yes

6. Review Comments to the Author

Reviewer #1: (No Response)

7. PLOS authors have the option to publish the peer review history of their article (what does this mean?). If published, this will include your full peer review and any attached files.

Reviewer #1: No

---

## [Editor Report · Acceptance letter]

PONE-D-25-64284R2

PLOS One

Dear Dr. Layton,

I'm pleased to inform you that your manuscript has been deemed suitable for publication in PLOS One. Congratulations! Your manuscript is now being handed over to our production team.

Kind regards,

on behalf of

Dr. Toshio Matsumoto

Academic Editor

PLOS One